# NeuronTune: Towards Self-Guided Spurious Bias Mitigation

**Guangtao Zheng** [1]   **Wenqian Ye** [1]   **Aidong Zhang** [1]

## Abstract

Deep neural networks often develop spurious bias, reliance on correlations between non-essential features and classes for predictions. For example, a model may identify objects based on frequently co-occurring backgrounds rather than intrinsic features, resulting in degraded performance on data lacking these correlations. Existing mitigation approaches typically depend on external annotations of spurious correlations, which may be difficult to obtain and are not relevant to the spurious bias in a model. In this paper, we take a step towards self-guided mitigation of spurious bias by proposing NeuronTune, a post hoc method that directly intervenes in a model's internal decision process. Our method probes in a model's latent embedding space to identify and regulate neurons that lead to spurious prediction behaviors. We theoretically justify our approach and show that it brings the model closer to an unbiased one. Unlike previous methods, NeuronTune operates without requiring spurious correlation annotations, making it a practical and effective tool for improving model robustness. Experiments across different architectures and data modalities demonstrate that our method significantly mitigates spurious bias in a self-guided way.

## 1. Introduction

Deep neural networks trained using empirical risk minimization (ERM) often develop spurious bias: a tendency to rely on spurious correlations for predictions. A spurious correlation refers to a non-causal relationship between a class and an attribute that is not essential for defining the class, commonly referred to as a spurious attribute (Ye et al., 2024). For example, the class of waterbird and the background of the water can form spurious correlations in the predictions of waterbird (Sagawa et al., 2019), as the background of the water is a spurious attribute. In contrast, core attributes, such as bird feathers, causally determine a class. A model with spurious bias may achieve a high prediction accuracy (Beery et al., 2018; Geirhos et al., 2019; 2020; Xiao et al., 2021; Zheng et al., 2024a) even without core attributes, such as identifying an object only by its frequently co-occurring background (Geirhos et al., 2020). However, the model may perform poorly on the data lacking the learned spurious correlations, which poses a great challenge to robust model generalization.

Existing methods (Sagawa et al., 2019; Kirichenko et al., 2023; Deng et al., 2024) that mitigate spurious bias are mostly at the sample level, using a curated set of samples with annotations of spurious correlations called *group labels* to retrain a biased model. A group label (class, spurious attribute) annotates a sample with a spurious attribute and its class label, representing a spurious correlation. However, group labels are difficult to acquire and often require costly human-guided annotations. To circumvent this, group label estimation (Nam et al., 2022) and various sample reweighting mechanisms (Nam et al., 2020; Liu et al., 2021; Kim et al., 2022; Qiu et al., 2023; LaBonte et al., 2024) are adopted using the idea that spurious bias can be identified through the misclassification of bias-conflicting samples.

Despite significant progress in spurious bias mitigation, existing sample-level methods that rely on group labels or sample reweighting offer limited and indirect control over how spurious bias is addressed. On the one hand, group labels are data annotations that are *external* to a model and may not accurately reflect the specific spurious bias developed in the model. On the other hand, sample reweighting does not directly target the internal mechanisms that give rise to spurious bias. This highlights the need for a **self-guided approach** that **directly intervenes in a model's decision process**, providing more targeted and model-relevant signals for mitigating spurious bias than sample-level approaches.

To this end, we focus on developing self-guided methods that directly analyze the internal prediction mechanism of a model to identify components of the model that are affected by spurious bias and then mitigate their influence to final predictions. We take a step towards this goal by proposing a novel method termed **NeuronTune**, which sys-

[1]Department of Computer Science, University of Virginia, Charlottesville, VA, USA. Correspondence to: Guangtao Zheng <gz5hp@virginia.edu>.

*Proceedings of the 42nd International Conference on Machine Learning*, Vancouver, Canada. PMLR 267, 2025. Copyright 2025 by the author(s).

tematically reduces spurious bias in deep neural networks. NeuronTune first probes in the latent embedding space of a trained model to identify dimensions (neurons) of sample embeddings *affected* by spurious bias, termed *biased dimensions*—those where spurious attributes predominantly contribute to prediction errors (Bykov et al., 2023; Singla & Feizi, 2021). Those dimensions can be identified when high activation magnitudes are strongly associated with incorrect predictions, indicating that features represented by those dimensions are not truly predictive of target classes. Importantly, rather than attempting to explicitly distinguish dimensions representing spurious and core attributes, an inherently challenging task given the complex entanglement of features in deep networks, NeuronTune instead identifies *biased dimensions* and suppresses the contributions of the these dimensions to final predictions. This intervention encourages the model to discover robust decision rules and mitigates spurious bias in the model.

Compared with the existing sample-level methods for spurious bias mitigation, NeuronTune provides direct intervention at the neuron level, allowing for more precise and targeted control over the mitigation of spurious bias during model tuning. Unlike approaches that rely on sample-level annotations such as group labels, NeuronTune enables the model to self-debias without external supervision. This makes it applicable in standard ERM training settings, where no additional annotations beyond class labels are available. As a result, NeuronTune serves as a practical and effective post hoc tool for mitigating spurious bias.

We theoretically demonstrate that neuron activations coupled with their final prediction outcomes provide self-identifying information on whether the neurons are affected by spurious bias. Our theoretical findings further suggest a practical metric for identifying biased dimensions and proves that NeuronTune can bring a model closer to the unbiased one. Experiments on vision and text datasets with different model architectures confirm the effectiveness of our method.

## 2. Related Work

Depending on the availability of external supervision, we summarize prior spurious bias mitigation methods into *supervised, semi-supervised, and unsupervised* categories.

**Supervised Spurious Bias Mitigation.** In this setting, certain spurious correlations in data are given in the form of group labels. With group labels in the training data, balancing the size of the groups (Cui et al., 2019; He & Garcia, 2009), upweighting groups that do not have specified spurious correlations (Byrd & Lipton, 2019), or optimizing the worst-group objective (Sagawa et al., 2019) can be effective. Regularization strategies, such as using information bottle-

neck (Tartaglione et al., 2021) or the distributional distance between bias-aligned samples (Barbano et al., 2023), are also proved to be effective in spurious bias mitigation. The concept of neural collapse has also been exploited recently for spurious bias mitigation (Wang et al., 2024). However, this setting requires to know what spurious bias needs to be mitigated *a priori* and only focuses on mitigating the specified spurious bias.

**Semi-Supervised Spurious Bias Mitigation.** This setting aims to mitigate spurious bias without extensive spurious correlation annotations. A small portion of group labels in a held-out set are required for achieving optimal performance. One line of works is to use data augmentation (Zhang et al., 2018; Han et al., 2022; Wu et al., 2023; Yao et al., 2022). Some methods propose to infer group labels via misclassified samples (Liu et al., 2021), clustering hidden embeddings (Zhang et al., 2022), or training a group label estimator (Nam et al., 2022). Creager et al. (2021) adopts invariant learning with inferred group labels. Other approaches include using biased models (Bahng et al., 2020), poisoning attack (Zhang et al., 2024), and mining intermediate attribute samples (Zhang et al., 2023). Recently, LaBonte et al. (2024) proposes to only use a small set of samples with group labels selected by the early-stop disagreement criterion. Relevant to our method are the last layer retraining (Kirichenko et al., 2023; Qiu et al., 2023) methods which only retrain the last layer of a model. In addition to retraining the last layer to minimize computational overhead, our method intervenes the internal decision process of a targeted model for more relevant and targeted spurious bias mitigation than existing methods.

**Unsupervised Spurious Bias Mitigation.** The goal in this setting is to train a robust model without using any group labels. Typically, we would expect relatively lower performance for methods working in this setting than in the above two settings as no information regarding the spurious correlations in test data is provided. Prior works use vision-language models to extract group labels (Zheng et al., 2024c;b), upweights training samples that are misclassified by a bias-amplified model (Li et al., 2024), or regularizes model retraining with detected prediction shortcuts in the latent space of a model (Zheng et al., 2025). A recent work (He et al., 2025) uses the observation that features with high confidence are likely to be spurious and mitigates spurious bias by erasing the corresponding activations. Our method considers the activation patterns of both correctly and incorrectly predicted samples of the same class to identify and mitigate spurious features.

## 3. Problem Setting

We consider a standard classification problem. The training set $\mathcal{D}_{\text{train}} = \{(\mathbf{x}, y) | \mathbf{x} \in \mathcal{X}, y \in \mathcal{Y}\}$ typically contains

data groups $\mathcal{D}_g^{\text{tr}}$ with $\mathcal{D}_{\text{train}} = \cup_{g \in \mathcal{G}} \mathcal{D}_g^{\text{tr}}$, where $\mathbf{x}$ denotes a sample in the input space $\mathcal{X}$, $y$ is the corresponding label in the finite label space $\mathcal{Y}$, $g := (y, a)$ denotes the group label defined by the combination of a class label $y$ and a spurious attribute $a \in \mathcal{A}$, where $\mathcal{A}$ denotes all spurious attributes in $\mathcal{D}_{\text{train}}$, and $\mathcal{G}$ denotes all possible group labels. Sample-label pairs in the group $\mathcal{D}_g^{\text{tr}}$ have the same class label $y$ and the same spurious attribute $a$.

**Our Scenario.** We consider *unsupervised spurious bias mitigation*, where no group labels are available, resembling a standard ERM training. A commonly used performance metric is the worst-group accuracy (WGA), which is the accuracy on the worst performing data group in the test set $\mathcal{D}_{\text{test}}$, i.e., WGA $= \min_{g \in \mathcal{G}} \text{Acc}(f, \mathcal{D}_g^{\text{te}})$, where $\mathcal{D}_g^{\text{te}}$ denotes a group of data in $\mathcal{D}_{\text{test}}$ with $\mathcal{D}_{\text{test}} = \cup_{g \in \mathcal{G}} \mathcal{D}_g^{\text{te}}$, and $f$ denotes a trained model. Typically, data in $\mathcal{D}_{\text{train}}$ is imbalanced across groups, and the model $f$ tends to favor certain data groups, resulting in a low WGA. Improving WGA without knowing group labels during training is challenging.

# 4. Methodology

We propose *NeuronTune*, a self-guided method for mitigating spurious bias without requiring group labels. NeuronTune identifies neurons (dimensions) affected by spurious bias in a model's latent space and tunes the model while suppressing the identified neurons. In Section 4.1, we present an analytical framework that outlines the design principles and theoretical properties of NeuronTune. Section 4.2 introduces a practical implementation for mitigating spurious bias in real-world settings.

## 4.1. NeuronTune: An Analytical Framework

At the core of NeuronTune is the identification of neurons that are affected by spurious bias. We establish an analytical framework to (1) elucidate the principle of neuron selection, (2) derive a selection metric that follows the principle of neuron selection, and (3) reveal the mechanism of NeuronTune in mitigating spurious bias.

### 4.1.1. DATA AND PREDICTION MODELS

**Data Model.** We design a data generation process that facilitates learning spurious correlations. Following the setting in (Arjovsky et al., 2019; Ye et al., 2023), we model a sample-label pair $(\mathbf{x}, y)$ in $\mathcal{D}_{\text{train}}$ as:

$$\mathbf{x} = \mathbf{x}_{\text{core}} \oplus \mathbf{x}_{\text{spu}} \in \mathbb{R}^{D \times 1}, \ y = \boldsymbol{\beta}^T \mathbf{x}_{\text{core}} + \varepsilon_{\text{core}}, \quad (1)$$

where $\mathbf{x}_{\text{core}} \in \mathbb{R}^{D_1 \times 1}$ is the core component, $\oplus$ denotes the vector concatenation operator, and the spurious component $\mathbf{x}_{\text{spu}} \in \mathbb{R}^{D_2 \times 1}$ with $D_1 + D_2 = D$ is associated with the label $y$ with the following relation:

$$\mathbf{x}_{\text{spu}} = (2a - 1)\boldsymbol{\gamma} y + \boldsymbol{\varepsilon}_{\text{spu}}, a \sim \text{Bern}(p), \quad (2)$$

where $(2a - 1) \in \{-1, +1\}$, $a \sim \text{Bern}(p)$ is a Bernoulli random variable, and $p$ is close to 1, indicating that $\mathbf{x}_{\text{spu}}$ is mostly predictive of $y$ but not always. In (1) and (2), $\boldsymbol{\beta} \in \mathbb{R}^{D_1 \times 1}$ and $\boldsymbol{\gamma} \in \mathbb{R}^{D_2 \times 1}$ are coefficients with unit $L^2$ norm, and $\varepsilon_{\text{core}} \in \mathbb{R}$ and $\boldsymbol{\varepsilon}_{\text{spu}} \in \mathbb{R}^{D_2 \times 1}$ represent the variations in the core and spurious components, respectively. We set $\varepsilon_{\text{core}}$ and each element in $\boldsymbol{\varepsilon}_{\text{spu}}$ as zero-mean Gaussian random variables with the variances $\eta_{\text{core}}^2$ and $\eta_{\text{spu}}^2$, respectively. We set $\eta_{\text{core}}^2 \gg \eta_{\text{spu}}^2$ to facilitate learning spurious correlations (Sagawa et al., 2019).

**Prediction Model.** We adopt a linear regression model with two linear layers (Ye et al., 2023) defined as $f(\mathbf{x}) = \mathbf{b}^T \mathbf{W} \mathbf{x}$, where $\mathbf{W} \in \mathbb{R}^{M \times D}$ denotes the embedding matrix simulating a feature extractor, $\mathbf{b} \in \mathbb{R}^{M \times 1}$ denotes the last layer, and $M$ is the number of embedding dimensions. The model $f(\mathbf{x})$ can be further expressed as follows,

$$\begin{aligned} f(\mathbf{x}) &= \sum_{i=1}^{M} b_i(\mathbf{x}_{\text{core}}^T \mathbf{w}_{\text{core},i} + \mathbf{x}_{\text{spu}}^T \mathbf{w}_{\text{spu},i}) \\ &= \mathbf{x}_{\text{core}}^T \mathbf{u}_{\text{core}} + \mathbf{x}_{\text{spu}}^T \mathbf{u}_{\text{spu}}, \end{aligned} \quad (3)$$

where $\mathbf{w}_{\text{core},i} \in \mathbb{R}^{D_1 \times 1}$, $\mathbf{w}_{\text{spu},i} \in \mathbb{R}^{D_2 \times 1}$, $\mathbf{w}_i^T = [\mathbf{w}_{\text{core},i}^T, \mathbf{w}_{\text{spu},i}^T] \in \mathbb{R}^{1 \times D}$ is the $i$-th row of $\mathbf{W}$, $\mathbf{u}_{\text{core}} = \sum_{i=1}^{M} b_i \mathbf{w}_{\text{core},i}$, and $\mathbf{u}_{\text{spu}} = \sum_{i=1}^{M} b_i \mathbf{w}_{\text{spu},i}$. The training objective is $\ell_{\text{tr}}(\mathbf{W}, \mathbf{b}) = \frac{1}{2}\mathbb{E}_{(\mathbf{x},y) \in \mathcal{D}_{\text{train}}} \|f(\mathbf{x}) - y\|_2^2$.

**Remark:** To better understand our data and prediction models, consider that $a$ in Eq. (2) controls subpopulations in data, e.g., when $a = 1$, it may represent a group of waterbirds on water, and when $a = 0$, it may represent a group of waterbirds on land. The probability $p$ controls the severity of imbalance in subpopulations. When $p$ is close to one, the data is severely imbalanced in subpopulations. After training with ERM, the model minimizes the training loss, i.e., maximizes the average-case accuracy, but obtains a large nonzero weight on the spurious feature (Lemma 1 in Appendix) and is away from the optimal model (Corollary 1 in Appendix). For example, the model may focus on correctly classifying waterbirds on water, at the expense of its ability to recognize waterbirds on land.

### 4.1.2. PRINCIPLE OF NEURON SELECTION

NeuronTune aims to identify neurons that reflect spurious bias. Proposition 4.1 specifies the principle of NeuronTune in terms of what neurons are to be identified and suppressed during model tuning.

**Proposition 4.1** (**Principle of NeuronTune**). *Given the model* $f(\mathbf{x}) = \mathbf{b}^T \mathbf{W} \mathbf{x}$ *trained with the data specified in* (1) *and* (2)*, it captures spurious correlations when* $\boldsymbol{\gamma}^T \mathbf{w}_{spu,i} < 0, i \in \{1, \ldots, M\}$. *The principle of NeuronTune is to suppress neurons containing negative* $\boldsymbol{\gamma}^T \mathbf{w}_{spu,i}$.

If $\boldsymbol{\gamma}^T \mathbf{w}_{\text{spu},i} \geq 0$, the model handles the spurious component

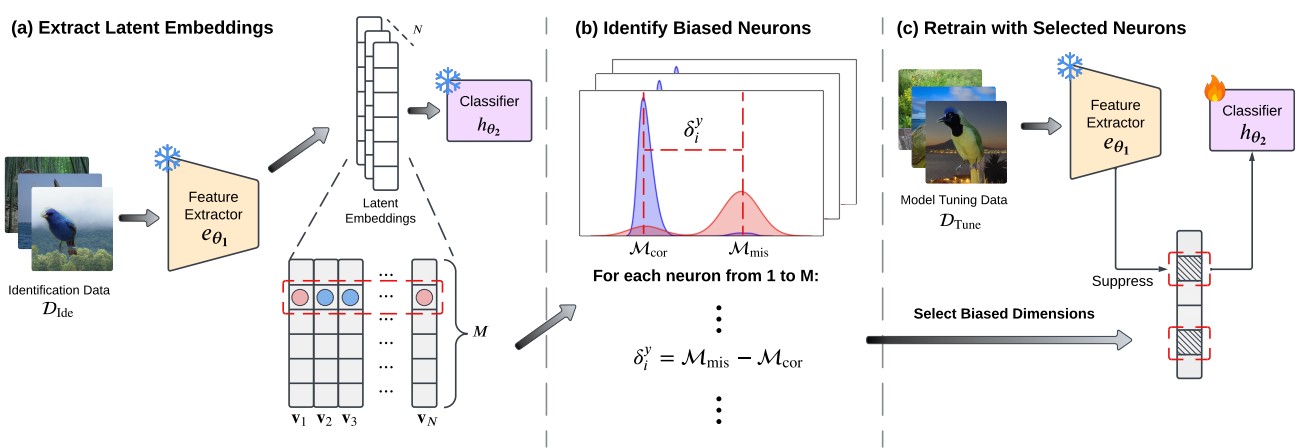

*Figure 1.* Practical implementation of NeuronTune. (a) Extract latent embeddings $\mathbf{v}_1, \ldots, \mathbf{v}_N$ and prediction outcomes (blue for correct and red for incorrect predictions) from an ERM-trained model using the identification data $\mathcal{D}_{\mathrm{Ide}}$. (b) Identify biased neurons (dimensions) utilizing the statistics $\mathcal{M}_{\mathrm{mis}}$ and $\mathcal{M}_{\mathrm{cor}}$ derived from neuron activations for correct (blue) and incorrect (red) predictions from Eq. (5). (c) Retrain the last prediction layer on $\mathcal{D}_{\mathrm{Tune}}$ while keeping the feature extractor frozen and suppressing identified biased dimensions.

correctly. Specifically, when $a = 1$, the spurious component $\mathbf{x}_{\mathrm{spu}}$ positively correlates with the core component $\mathbf{x}_{\mathrm{core}}$ and contributes to the output, whereas when $a = 0$, its correlation with $\mathbf{x}_{\mathrm{core}}$ breaks with a negative one and has a negative contribution to the output. The relations reverse when $\gamma^T \mathbf{w}_{\mathrm{spu},i} < 0$, i.e., the model utilizes $\mathbf{x}_{\mathrm{spu}}$ even when the correlation breaks, demonstrating a strong reliance on the spurious component instead of the core component. The proof is in Appendix A.2.4.

#### 4.1.3. METRIC FOR NEURON SELECTION

Guided by the principle of NeuronTune in Proposition 4.1, the following theorem gives a practical metric to select neurons that are affected by spurious bias.

**Theorem 4.2** (**Metric for Neuron Selection**). *Given the model $f(\mathbf{x}) = \mathbf{b}^T \mathbf{W} \mathbf{x}$, we cast it to a classification model by training it to regress $y \in \{-\mu, \mu\}$ ($\mu > 0$) on $\mathbf{x}$ based on the data model specified in* (1) *and* (2)*, where $\mu = \mathbb{E}[\boldsymbol{\beta}^T \mathbf{x}_{core}]$. The metric $\delta_i^y$ defined in the following can identify neurons affected by spurious bias when $\delta_i^y > 0$:*

$$\delta_i^y = Med(\bar{\mathcal{V}}_i^y) - Med(\hat{\mathcal{V}}_i^y),$$

*where $\bar{\mathcal{V}}_i^y$ and $\hat{\mathcal{V}}_i^y$ are the sets of activation values for misclassified and correctly predicted samples with the label $y$ from the $i$-th neuron, respectively; an activation value is defined as $\mathbf{x}_{core,i}^T \mathbf{w}_{core,i} + \mathbf{x}_{spu}^T \mathbf{w}_{spu,i}$, and $Med(\cdot)$ returns the median of an input set of values.*

We show in Appendix A.2.5 that the theorem establishes the approximation $\delta_i^y \approx -2\mu \gamma^T \mathbf{w}_{\mathrm{spu},i}$, which confirms that neurons selected by the metric defined above follow the principle in Proposition 4.1. Adopting medians in the metric makes the metric robust to outlier values.

Let $\mathcal{M}_{\mathrm{mis}} = \mathrm{Med}(\bar{\mathcal{V}}_i^y)$ and $\mathcal{M}_{\mathrm{cor}} = \mathrm{Med}(\hat{\mathcal{V}}_i^y)$. Intuitively, a high $\mathcal{M}_{\mathrm{mis}}$ indicates that high activations at the $i$-th dimension contribute to misclassification when predicting the class $y$. A low $\mathcal{M}_{\mathrm{cor}}$ implies that the $i$-th dimension has little effect in correctly predicting the class $y$. Thus, a large difference between $\mathcal{M}_{\mathrm{mis}}$ and $\mathcal{M}_{\mathrm{cor}}$, i.e., a large $\delta_i^y$, indicates that the $i$-th dimension represents features that are irrelevant to the class $y$. In other words, with a high likelihood, the dimension is affected by spurious bias. In contrast, a negative $\delta_i^y$ highlights the relevance of the $i$-th dimension for predictions as most correctly predicted samples have high activation values in this dimension, and most incorrectly predicted samples have low activation values.

**Remark:** Proposition 4.1 and Theorem 4.2 state that when a spurious correlation breaks, neurons that continue to positively contribute to mispredictions will be selected. For example, in the case of waterbird with water and land backgrounds, neurons that cause misclassification on images of waterbird appearing on land will be identified.

#### 4.1.4. MECHANISM OF NEURONTUNE

NeuronTune mitigates spurious bias by retraining the last layer while suppressing (zeroing out) the identified neurons. The following theorem shows that this improves model robustness and explains how it achieves this.

**Theorem 4.3** (**NeuronTune Mitigates Spurious Bias**). *Consider the model $f^*(\mathbf{x}) = \mathbf{x}^T \mathbf{u}^*$ trained on the biased training data with $p \gg 0.5$, where $\mathbf{u}^{*T} = [\mathbf{u}_{core}^{*T}, \mathbf{u}_{spu}^{*T}]$. Under the mild assumption that $\boldsymbol{\beta}^T \mathbf{w}_{core,i} \approx \gamma^T \mathbf{w}_{spu,i}, \forall i = 1, \ldots, M$, then applying NeuronTune to $f^*(\mathbf{x})$ produces a model that is closer to the unbiased one.*

The assumption $\boldsymbol{\beta}^T \mathbf{w}_{\text{core},i} \approx \boldsymbol{\gamma}^T \mathbf{w}_{\text{spu},i}, \forall i = 1, \ldots, M$ generally holds for a biased model, as the model has learned to associate spurious attributes with core attributes. The proof is in Appendix A.2.6. Denote the NeuronTune solution by $\mathbf{u}_{\text{core}}^\dagger$ and $\mathbf{u}_{\text{spu}}^\dagger$. Our finding reveals that retraining the last layer does not alter the weight on the spurious component, i.e., $\mathbf{u}_{\text{spu}}^\dagger = \mathbf{u}_{\text{spu}}^*$, which is the optimal solution achievable by last-layer retraining methods (see Lemma 3 in Appendix). However, it does adjust $\mathbf{u}_{\text{core}}^\dagger$ to be closer to the optimal weight on the core component, $\boldsymbol{\beta}$. Overall, NeuronTune brings the model parameters closer to the optimal, unbiased solution compared to the parameters of the original biased model. Therefore, NeuronTune is guaranteed to outperform the ERM-trained model. Further discussion on the connection to last-layer retraining methods is provided in Appendix A.3.

**Remark:** Our findings suggest that our approach makes a slight trade-off in average-case accuracy to achieve improved worst-group accuracy. For example, our method may slightly reduce the model's ability to classify waterbird on water due to a *relative* decrease in reliance on the water feature, while significantly enhancing its ability to classify waterbird on land.

### 4.2. NeuronTune: Practical Implementation

For real-world spurious bias mitigation, we consider a well-trained ERM model $f_{\boldsymbol{\theta}}$ where $\boldsymbol{\theta} = \arg\min_{\boldsymbol{\theta}'} \mathbb{E}_{(\mathbf{x},y)\in\mathcal{D}_{\text{train}}} \ell(f_{\boldsymbol{\theta}'}(\mathbf{x}), y)$, and $\ell$ denotes the cross-entropy loss function. The model $f_{\boldsymbol{\theta}} = e_{\boldsymbol{\theta}_1} \circ h_{\boldsymbol{\theta}_2}$ consists of a feature extractor $e_{\boldsymbol{\theta}_1} : \mathcal{X} \to \mathbb{R}^M$ followed by a linear classifier $h_{\boldsymbol{\theta}_2} : \mathbb{R}^M \to \mathbb{R}^{|\mathcal{Y}|}$, where $M$ is the number of dimensions of latent embeddings obtained from $e_{\boldsymbol{\theta}_1}$, $\circ$ denotes the function composition operator, and $\boldsymbol{\theta} = \boldsymbol{\theta}_1 \cup \boldsymbol{\theta}_2$.

NeuronTune aligns best with our theoretical analysis when implemented as a last-layer retraining method where the feature extractor $e_{\boldsymbol{\theta}_1}$ is fixed and the last layer is linear and tunable. Fig. 1 gives an overview of NeuronTune which mainly includes identifying affected neurons (Section 4.2.1) and model tuning with identified neurons (Section 4.2.2).

#### 4.2.1. IDENTIFYING AFFECTED NEURONS

As shown in Fig. 1(a), we use a set of identification data $\mathcal{D}_{\text{Ide}}$, which typically contains a set of diverse features not seen by the model, to identify dimensions (neurons) affected by spurious bias in the model's latent space. We first extract latent embeddings and prediction outcomes for samples of class $y$ in $\mathcal{D}_{\text{Ide}}$, i.e.,

$$\mathcal{V}^y = \{(\mathbf{v}, o) | \mathbf{v} = e_{\boldsymbol{\theta}_1}(\mathbf{x}), \forall (\mathbf{x}, y) \in \mathcal{D}_{\text{Ide}}\}, \quad (4)$$

where $o = \mathbb{1}\{\arg\max f_{\boldsymbol{\theta}}(\mathbf{x}) == y\}$, $\mathbf{v} \in \mathbb{R}^M$ is an $M$-dimensional latent embedding of $\mathbf{x}$, and $o$ is the corresponding prediction outcome with $\mathbb{1}$ being an indicator function.

**Identification Criterion.** As shown in Fig. 1(b), for each embedding dimension $i$, we separate $\mathcal{V}^y$ into two sets $\hat{\mathcal{V}}_i^y$ and $\bar{\mathcal{V}}_i^y$, representing values at the $i$-th embedding dimension from $\mathcal{V}^y$, contributing respectively to correct and incorrect predictions, i.e., $\hat{\mathcal{V}}_i^y = \{\mathbf{v}[i] | (\mathbf{v}, 1) \in \mathcal{V}^y\}$, and $\bar{\mathcal{V}}_i^y = \{\mathbf{v}[i] | (\mathbf{v}, 0) \in \mathcal{V}^y\}, \forall i = 1, \ldots, M, y \in \mathcal{Y}$, where $\mathbf{v}[i]$ denotes the $i$-th dimension of $\mathbf{v}$. We propose a *spuriousness score* $\delta_i^y$ to measure the spuriousness of the $i$-th dimension when predicting the class $y$. Following the insight from Theorem 4.2, we define $\delta_i^y$ as follows:

$$\delta_i^y = \mathcal{M}_{\text{mis}} - \mathcal{M}_{\text{cor}}, \quad (5)$$

where $\mathcal{M}_{\text{mis}} = \text{Med}(\bar{\mathcal{V}}_i^y)$ and $\mathcal{M}_{\text{cor}} = \text{Med}(\hat{\mathcal{V}}_i^y)$.

Theorem 4.2 assumes that each dimension of input embeddings consists of a linear combination of spurious and core components. While it generally holds that each dimension represents a mixture of spurious and core components, in real-world scenarios, the combination is typically nonlinear. To account for this, we introduce $\lambda$ as a threshold and identify dimensions using the following criterion:

$$\mathcal{S} = \{i | \delta_i^y > \lambda, \forall i = 1, \ldots, M, y \in \mathcal{Y}\}. \quad (6)$$

We set $\lambda$ to 0 by default, as it works well in practice.

In the following, we refer to a dimension as a **biased dimension** when $\delta_i^y > \lambda$ and **unbiased dimension** otherwise. A biased (unbiased) dimension does not imply that the dimension exclusively represents spurious (core) attributes. In practice, an unbiased dimension exhibits high activation values for target classes, whereas a biased dimension shows high activation values for undesired classes. Visualizations of several identified biased and unbiased dimensions on real-world datasets are provided in Appendix A.9.

We include the dimensions identified for all the classes into the set $\mathcal{S}$ since an identified biased dimension for one class cannot serve as a core contributor to predicting some other class in a well-defined classification task. For example, consider that the dimension representing "blue color" is biased for the "rectangle" class while being unbiased for the "blue color" class. This happens when we have a blue rectangle as the input, which makes the classification ambiguous.

#### 4.2.2. MODEL TUNING WITH IDENTIFIED NEURONS

As illustrated in Fig. 1(c), we tune the last prediction layer while suppressing the signals from the identified biased dimensions. In this way, we explicitly intervene the internal decision process of the model to discover robust decision rules beyond using spurious correlations.

**Learning Objective.** Concretely, given a model tuning dataset $\mathcal{D}_{\text{Tune}}$, we optimize the following objective,

$$\boldsymbol{\theta}_2^* = \arg\min_{\boldsymbol{\theta}_2} \mathbb{E}_{\mathcal{B} \sim \mathcal{D}_{\text{Tune}}} \mathbb{E}_{(x,y) \in \mathcal{B}} \ell(h_{\boldsymbol{\theta}_2}(\tilde{\mathbf{v}}), y), \quad (7)$$

where $\mathcal{B}$ contains *class-balanced* sample-label pairs from $\mathcal{D}_{\text{Tune}}$, addressing that the classifier may favor certain classes during model tuning, and $\tilde{\mathbf{v}}$ is the latent embedding after zeroing-out activations on the biased dimensions in $\mathcal{S}$. Unless otherwise stated, we use $\mathcal{D}_{\text{train}}$ as $\mathcal{D}_{\text{Tune}}$.

**Model Selection.** Without group labels, it is challenging to select robust models (Liu et al., 2021; Yang et al., 2023). We address this by designing a novel model selection metric, termed *spuriousness fitness score (SFit)*, which is the sum of magnitudes of spuriousness scores across dimensions and classes, i.e., $\text{SFit} = \sum_{m=1}^{M} \sum_{y \in \mathcal{Y}} \text{Abs}(\delta_m^y)$, where $\text{Abs}(\cdot)$ returns the absolute value of a given input. The score holistically summarizes whether biased and unbiased dimensions in the model are distinguishable. A low SFit indicates that the model tends to memorize samples. Empirically, we find that a high SFit effectively selects a robust model.

NeuronTune is highly efficient as it only requires tuning the last layer of the model. We use (6) and (7) to iteratively perform the biased dimension detection and model tuning while using SFit for model selection.

# 5. Experiments

## 5.1. Datasets

We tested NeuronTune on four image datasets and two text datasets, each with different types of spurious attributes. (1) **Waterbirds** (Sagawa et al., 2019) is an image dataset for recognizing waterbird and landbird. It is generated synthetically by combining images of the two bird types from the CUB dataset (Welinder et al., 2010) and the backgrounds, water and land, from the Places dataset (Zhou et al., 2017). (2) **CelebA** (Liu et al., 2015) is a large-scale image dataset of celebrity faces. The task is to identify hair color, non-blond or blond, with male or female as the spurious attribute. (3) **ImageNet-9** (Xiao et al., 2021) is a subset of ImageNet (Deng et al., 2009) containing nine super-classes. It comprises images with different background and foreground signals and can be used to assess how much models rely on image backgrounds. (4) **ImageNet-A** (Hendrycks et al., 2021) is a dataset of real-world images, adversarially curated to test the limits of classifiers such as ResNet-50. We used this dataset to test the robustness of a classifier after training it on ImageNet-9. (5) **MultiNLI** (Williams et al., 2018) is a text classification dataset with three classes: neutral, contradiction, and entailment, representing the natural language inference relationship between a premise and a hypothesis. The spurious attribute is the presence of negation. (6) **CivilComments** (Borkan et al., 2019) is a binary text classification dataset aimed at predicting whether an internet comment contains toxic language. The spurious attribute involves references to eight demographic identities. The dataset uses standard splits provided by the WILDS

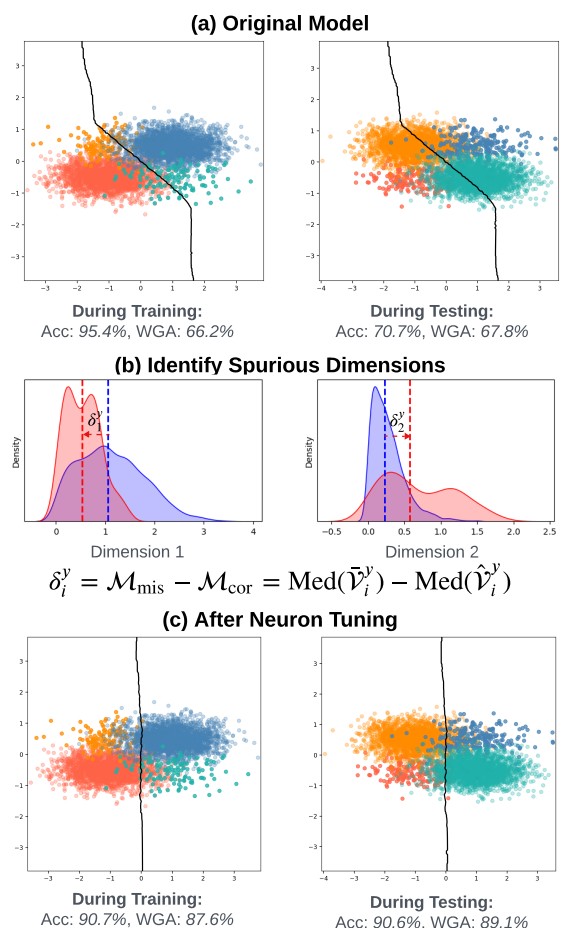

*Figure 2.* Synthetic experiment. (a) Training and test data distributions along with the decision boundaries of the trained model. (b) Value distributions of the correctly (blue) and incorrectly (red) predicted samples at the first (left) and second (right) dimensions of input embeddings, with the second dimension identified as a biased dimension. (c) NeuronTune improves WGA. Data groups $(y = +1, a = 1)$: red dots; $(y = +1, a = 0)$: orange dots; $(y = -1, a = 0)$: blue dots; $(y = -1, a = 1)$: green dots.

benchmark (Koh et al., 2021).

## 5.2. Experimental Setup

**Training Details.** We first trained ERM models on each of the datasets. We used ResNet-50 and ResNet-18 (He et al., 2016) models pretrained on ImageNet for experiments on the Waterbirds and CelebA datasets, and on the ImageNet-9 and ImageNet-A datasets, respectively. For text datasets, we used the BERT model (Kenton & Toutanova, 2019) pretrained on Book Corpus and English Wikipedia data. We followed the settings in Izmailov et al. (2022) for ERM training, with the best models selected based on the average validation accuracy. For our NeuronTune training, unless otherwise stated, we used the validation data

*Table 1.* Comparison of worst-group accuracy (%), average accuracy (%), and accuracy gap (%) on the image datasets. $^\dagger$ denotes using a fraction of validation data for model tuning. The best result in each group of methods is in **boldface**.

| Algorithm | Group annotations | | Waterbirds | | | CelebA | | |
|---|---|---|---|---|---|---|---|---|
| | Train | Val | WGA ($\uparrow$) | Acc. ($\uparrow$) | Acc. Gap ($\downarrow$) | WGA ($\uparrow$) | Acc. ($\uparrow$) | Acc. Gap ($\downarrow$) |
| ERM (Vapnik, 1999) | - | - | 72.6 | 97.3 | 24.7 | 47.2 | 95.6 | 48.4 |
| JTT (Liu et al., 2021) | No | Yes | 86.7 | 93.3 | 6.6 | 81.1 | 88.0 | 6.9 |
| SELF$^\dagger$ (LaBonte et al., 2024) | No | Yes | **93.0**$_{\pm0.3}$ | 94.0$_{\pm1.7}$ | **1.0** | 83.9$_{\pm0.9}$ | 91.7$_{\pm0.4}$ | 7.8 |
| CNC (Zhang et al., 2022) | No | Yes | 88.5$_{\pm0.3}$ | 90.9$_{\pm0.1}$ | 2.4 | **88.8**$_{\pm0.9}$ | 89.9$_{\pm0.5}$ | **1.1** |
| BAM (Li et al., 2024) | No | Yes | 89.2$_{\pm0.3}$ | 91.4$_{\pm0.4}$ | 2.2 | 83.5$_{\pm0.9}$ | 88.0$_{\pm0.4}$ | 4.5 |
| AFR (Qiu et al., 2023) | No | Yes | 90.4$_{\pm1.1}$ | 94.2$_{\pm1.2}$ | 3.8 | 82.0$_{\pm0.5}$ | 91.3$_{\pm0.3}$ | 9.3 |
| DFR$^\dagger$ (Kirichenko et al., 2023) | No | Yes | 92.4$_{\pm0.9}$ | 94.9$_{\pm0.3}$ | 2.5 | 87.0$_{\pm1.1}$ | 92.6$_{\pm0.5}$ | 5.6 |
| BPA (Seo et al., 2022) | No | No | 71.4 | - | - | 82.5 | - | - |
| GEORGE (Sohoni et al., 2020) | No | No | 76.2 | 95.7 | 19.5 | 52.4 | 94.8 | 42.4 |
| BAM (Li et al., 2024) | No | No | 89.1$_{\pm0.2}$ | 91.4$_{\pm0.3}$ | 2.3 | 80.1$_{\pm3.3}$ | 88.4$_{\pm2.3}$ | 8.3 |
| NeuronTune | No | No | 92.2$_{\pm0.3}$ | 94.4$_{\pm0.2}$ | 2.2 | 83.1$_{\pm1.1}$ | 92.0$_{\pm0.5}$ | 8.9 |
| NeuronTune$^\dagger$ | No | No | **92.5**$_{\pm0.9}$ | 94.5$_{\pm0.3}$ | **2.0** | **87.3**$_{\pm0.4}$ | 90.3$_{\pm0.5}$ | **3.0** |

as $\mathcal{D}_{\text{Ide}}$ and the training data as $\mathcal{D}_{\text{Tune}}$. We took the absolute values of neuron activations before the identification process, ensuring that high activation magnitudes reflect strong contributions to predictions. We ran the training under five different random seeds and reported average accuracies along with standard deviations. We provide full training details in Appendix A.8. Code is available at https://github.com/gtzheng/NeuronTune.

**Evaluation Metrics.** To evaluate the robustness to spurious bias, we adopt the widely accepted robustness metric, *worst-group accuracy (WGA)*, that gives the lower-bound performance of a classifier on the test set with various dataset biases. We also focus on the *accuracy gap* between the standard average accuracy and the worst-group accuracy as a measure of a classifier's reliance on spurious correlations. A high worst-group accuracy and a low accuracy gap indicate that the classifier is robust to spurious correlations and can fairly predict samples from different groups.

### 5.3. Synthetic Experiment

We considered an input $\mathbf{v} = [v^c, v^s, v^\epsilon] \in \mathbb{R}^3$ that has three dimensions: a core dimension with the core component $v^c \in \mathbb{R}$, a spurious dimension with the spurious component $v^s \in \mathbb{R}$, and a noise dimension with the noise component $v^\epsilon$. We generated training and test sets with sample-label pairs $(\mathbf{v}, y)$, where $y \in \{-1, +1\}$. The core component in $\mathbf{v}$ is a noisy version of the label $y$ in both sets. The spurious component in the training set is a noisy version of the spurious attribute $a = 0$ in 95% (5% for $a = 1$) of samples with $y = -1$ and in 5% (95% for $a = 1$) of samples with $y = +1$. The noise component is an independent zero-mean Gaussian variable. In the test set, for each label, we reduced the 95% group to 10%, effectively reversing the majority and minority group roles. We adopted a logistic

regression model $\phi_{\tilde{\mathbf{w}}}(\mathbf{v}) = 1/(1+\exp\{-(\mathbf{w}^T\mathbf{v}+b)\})$ with $\tilde{\mathbf{w}} = [\mathbf{w}, b]$. The model predicts $+1$ when $\phi_{\tilde{\mathbf{w}}}(\mathbf{v}) > 0.5$ and $-1$ otherwise. We trained $\phi_{\tilde{\mathbf{w}}}$ on the generated training data and tested it on the corresponding test data. Details of the data generation are provided in Appendix A.1.

Fig. 2 illustrates spurious bias and how NeuronTune mitigates it. First, we observe that the decision boundary of the trained model tends to separate the majority groups of training samples. This leads to a high average accuracy but a small WGA on the training set (Fig. 2(a), left) and poor performance on the test set (Fig. 2(a), right). Then, Fig. 2(b) demonstrates the value distributions of the first (core) and second (spurious) dimensions of the input samples with $y = -1$. NeuronTune identified the second dimension as a biased dimension, which indeed represents spurious attributes. Next, Fig. 2(c) shows that NeuronTune significantly improves WGA on both the training and test sets by suppressing the contributions from biased dimensions. Finally, independent of how NeuronTune works, there exists a tradeoff between average accuracy and WGA due to complexity of input samples, as demonstrated in the left parts of Figs. 2(a) and 2(c).

### 5.4. Comparison with Existing Approaches

We evaluated NeuronTune on both image and text datasets to showcase its effectiveness and versatility in handling different data modalities and model architectures. Our primary comparisons were with methods specifically designed for unsupervised spurious bias mitigation, where no group labels are available for bias mitigation. To provide additional context, we also included methods for semi-supervised spurious bias mitigation, which leverage group labels in the validation set to select robust models.

Results in the lower parts of Tables 1 and 2 were obtained

*Table 2.* Comparison of worst-group accuracy (%), average accuracy (%), and accuracy gap (%) on the text datasets. [†] denotes using a fraction of validation data for model tuning. The best result in each group of methods is in **boldface**.

| Algorithm | Group annotations | | MultiNLI | | | CivilComments | | |
|---|---|---|---|---|---|---|---|---|
| | Train | Val | WGA ($\uparrow$) | Acc. ($\uparrow$) | Acc. Gap ($\downarrow$) | WGA ($\uparrow$) | Acc. ($\uparrow$) | Acc. Gap ($\downarrow$) |
| ERM (Vapnik, 1999) | - | - | 67.9 | **82.4** | 14.5 | 57.4 | **92.6** | 35.2 |
| JTT (Liu et al., 2021) | No | Yes | 72.6 | 78.6 | **6.0** | 69.3 | 91.1 | 21.8 |
| SELF[†] (LaBonte et al., 2024) | No | Yes | $70.7_{\pm2.5}$ | $81.2_{\pm0.7}$ | 10.5 | $79.1_{\pm2.1}$ | $87.7_{\pm0.6}$ | 8.6 |
| CNC (Zhang et al., 2022) | No | Yes | - | - | - | $68.9_{\pm2.1}$ | $81.7_{\pm0.5}$ | 12.8 |
| BAM (Li et al., 2024) | No | Yes | $71.2_{\pm1.6}$ | $79.6_{\pm1.1}$ | 8.4 | $79.3_{\pm2.7}$ | $88.3_{\pm0.8}$ | 9.0 |
| AFR (Qiu et al., 2023) | No | Yes | $\mathbf{73.4}_{\pm0.6}$ | $81.4_{\pm0.2}$ | 8.0 | $68.7_{\pm0.6}$ | $89.8_{\pm0.6}$ | 21.1 |
| DFR[†] (Kirichenko et al., 2023) | No | Yes | $70.8_{\pm0.8}$ | $81.7_{\pm0.2}$ | 10.9 | $\mathbf{81.8}_{\pm1.6}$ | $87.5_{\pm0.2}$ | **5.7** |
| BAM (Li et al., 2024) | No | No | $70.8_{\pm1.5}$ | $80.3_{\pm1.0}$ | 9.5 | $79.3_{\pm2.7}$ | $88.3_{\pm0.8}$ | 9.0 |
| NeuronTune | No | No | $72.1_{\pm0.1}$ | $81.1_{\pm0.6}$ | 9.0 | $82.4_{\pm0.2}$ | $89.2_{\pm0.1}$ | 6.8 |
| NeuronTune[†] | No | No | $\mathbf{72.5}_{\pm0.3}$ | $80.3_{\pm0.6}$ | **7.8** | $\mathbf{82.7}_{\pm0.4}$ | $89.4_{\pm0.2}$ | **6.7** |

*Table 3.* Average accuracy (%) and accuracy gap (%) comparison on the ImageNet-9 and ImageNet-A datasets. All methods used ResNet-18 as the backbone. The best results are in **boldface**.

| Method | ImageNet-9 | ImageNet-A | Acc. Gap ($\downarrow$) |
|---|---|---|---|
| ERM (Vapnik, 1999) | $90.8_{\pm0.6}$ | $24.9_{\pm1.1}$ | 65.9 |
| StylisedIN (Geirhos et al., 2019) | $88.4_{\pm0.5}$ | $24.6_{\pm1.4}$ | 63.8 |
| RUBi (Cadene et al., 2019) | $90.5_{\pm0.3}$ | $27.7_{\pm2.1}$ | 62.8 |
| ReBias (Bahng et al., 2020) | $91.9_{\pm1.7}$ | $29.6_{\pm1.6}$ | 62.3 |
| LfF (Nam et al., 2020) | 86.0 | 24.6 | 61.4 |
| CaaM (Wang et al., 2021) | **95.7** | 32.8 | 62.9 |
| SSL+ERM (Kim et al., 2022) | $94.2_{\pm0.1}$ | $34.2_{\pm0.5}$ | 60.0 |
| LWBC (Kim et al., 2022) | $94.0_{\pm0.2}$ | $36.0_{\pm0.5}$ | 58.0 |
| **NeuronTune** | $93.7_{\pm0.1}$ | $\mathbf{37.3}_{\pm0.5}$ | **56.4** |

in the unsupervised spurious bias mitigation setting. In this setting, our method achieves the highest worst-group accuracies and smallest accuracy gaps across the datasets, highlighting its effectiveness in enhancing models' robustness to spurious bias and balancing performance across different data groups. Results in the upper parts of Tables 1 and 2 were from methods in the semi-supervised spurious bias mitigation setting. Methods in this setting benefit from group labels for selecting robust models. Despite this advantage, NeuronTune demonstrates strong self-debiasing capabilities, competing favorably with methods such as AFR and DFR that rely on group labels. When a half of the validation set was used in training, NeuronTune achieved better WGAs and accuracy gaps on three out of four datasets than DFR and SELF that exploited the same set of data for training.

Notably, compared with sample-level last-layer retraining methods, such as AFR, NeuronTune manipulates the neurons within a model, providing more targeted control on how spurious bias is mitigated. Hence, NeuronTune in theory can achieve better robustness to spurious bias (Appendix A.3). In general, NeuronTune compares favorably with AFR in terms of WGA and accuracy gap, with larger gains

achieved when AFR models were selected without group labels (Appendix A.4).

We further used the ImageNet-9 (Kim et al., 2022; Bahng et al., 2020) and ImageNet-A (Hendrycks et al., 2021) datasets to evaluate NeuronTune's robustness to distribution shifts, which are challenging to depict in group labels. We first trained an ERM model from scratch using ImageNet-9 and then fine-tuned its last layer with Neuron-Tune. In Table 3, NeuronTune achieves the best accuracy on the challenging ImageNet-A dataset, which is known for its natural adversarial examples. While this improvement comes with a slight trade-off in in-distribution accuracy on ImageNet-9, NeuronTune maintains the smallest accuracy gap between the two datasets, making it a robust method for out-of-distribution generalization.

Finally, in Tables 1, 2, and 3, we observe a common trade-off between average accuracy and WGA that exists across many spurious bias mitigation methods. For NeuronTune, this trade-off primarily occurs when samples sharing the same spurious attribute but belonging to different classes are difficult to separate in the latent space, as illustrated in

*Table 4.* Comparison of worst-group accuracy (%) between different choices of $\mathcal{D}_{\text{Ide}}$ and $\mathcal{D}_{\text{Tune}}$ as well as neuron-based tuning (NT) on the four datasets. The best results are in **boldface**.

| $\mathcal{D}_{\text{Ide}}$ | $\mathcal{D}_{\text{Tune}}$ | NT | Waterbirds | CelebA | MultiNLI | CivilComments |
|---|---|---|---|---|---|---|
| $\mathcal{D}_{\text{train}}$ | $\mathcal{D}_{\text{train}}$ | Yes | $78.0_{\pm 2.3}$ | $58.5_{\pm 1.2}$ | $42.0_{\pm 10.5}$ | $80.0_{\pm 10.5}$ |
| $\mathcal{D}_{\text{val}}$ | $\mathcal{D}_{\text{train}}$ | Yes | $92.2_{\pm 0.3}$ | $83.1_{\pm 1.1}$ | $72.1_{\pm 0.1}$ | $82.4_{\pm 0.2}$ |
| $\mathcal{D}_{\text{val}}$ | $\mathcal{D}_{\text{train}}$ | No | $82.7_{\pm 0.4}$ | $53.9_{\pm 0.0}$ | $63.4_{\pm 0.7}$ | $81.5_{\pm 0.5}$ |
| $\mathcal{D}_{\text{val}}/2$ | $\mathcal{D}_{\text{val}}/2$ | Yes | $\mathbf{92.5}_{\pm 0.9}$ | $\mathbf{87.3}_{\pm 0.4}$ | $\mathbf{72.5}_{\pm 0.3}$ | $\mathbf{82.7}_{\pm 0.4}$ |

*Table 5.* Analysis of the impact of partial suppression (masking value $> 0$) and full suppression (masking value $= 0$) on the performance of NeuronTune[†], evaluated on the CelebA dataset.

| Masking value | 0 | 0.2 | 0.4 | 0.6 | 0.8 | 1.0 |
|---|---|---|---|---|---|---|
| WGA ($\uparrow$) | $87.3_{\pm 0.4}$ | $71.5_{\pm 1.5}$ | $72.2_{\pm 1.2}$ | $72.9_{\pm 1.5}$ | $73.1_{\pm 1.5}$ | $73.0_{\pm 1.2}$ |
| Acc. ($\uparrow$) | $90.3_{\pm 0.5}$ | $93.8_{\pm 0.2}$ | $93.8_{\pm 0.3}$ | $93.8_{\pm 0.2}$ | $93.8_{\pm 0.2}$ | $93.9_{\pm 0.2}$ |

Fig. 2. While improving sample embeddings could help alleviate this issue, it often demands substantial computational resources. In contrast, NeuronTune, as a post hoc method, efficiently mitigates spurious bias by tuning only the last layer with low computational complexity (Appendix A.5) while still achieving a favorable balance between WGA and overall performance.

### 5.5. Ablation Studies

In Table 4, we compare NeuronTune's performance between different choices of the identification dataset $\mathcal{D}_{\text{Ide}}$ and the model tuning dataset $\mathcal{D}_{\text{Tune}}$. Additionally, we demonstrate the effectiveness of neuron-based tuning on the identified biased dimensions (denoted as NT).

When using $\mathcal{D}_{\text{Ide}} = \mathcal{D}_{\text{train}}$, we observe a relatively low performance across datasets. After switching to a held-out validation data $\mathcal{D}_{\text{val}}$, we observe significant performance improvements. This highlights the advantage of using a new and independent dataset to identify biased dimensions, as models may have already memorized patterns in $\mathcal{D}_{\text{train}}$. By default, NeuronTune adopts $\mathcal{D}_{\text{val}}$ as $\mathcal{D}_{\text{Ide}}$. It is important to note that using $\mathcal{D}_{\text{val}}$ to identify biased dimensions is analogous to using it for model selection. Hence, $\mathcal{D}_{\text{val}}$ is not directly used for updating model weights.

Next, we disabled NT during model tuning (NT=No), which effectively reduces NeuronTune to class-balanced model tuning. We observe consistent performance degradation across the four datasets, which validates the effectiveness of NT across datasets.

Moreover, inspired by the success of DFR (Kirichenko et al., 2023), which uses a half of the validation data for model tuning, we divided $\mathcal{D}_{\text{val}}$ into two equal halves: one half (denoted as $\mathcal{D}_{\text{val}}/2$) was used as $\mathcal{D}_{\text{Ide}}$, while the other half served as $\mathcal{D}_{\text{Tune}}$. Unlike DFR, our method does not rely on group labels in the validation data. This strategy leads to further performance improvements on datasets such as CelebA and MultiNLI, demonstrating the advantage of using separate and independent datasets for bias identification and model tuning. Identifying the optimal choice for $\mathcal{D}_{\text{Ide}}$ and $\mathcal{D}_{\text{Tune}}$ remains an avenue for future research.

Finally, we analyze different strategies for handling the identified biased dimensions, as shown in Table 5. Our default approach, described in Section 4.2.2, fully suppresses the activations on the biased dimensions by multiplying the activations with a masking value of zero. To explore the effect of partial suppression, we varied the masking value from 0.2 to 1.0, where 1.0 corresponds to no suppression. As shown in Table 5, on the CelebA dataset, only the full suppression strategy (masking value = 0) led to an improvement in WGA. This highlights that while partial suppression may reduce the loss in average accuracy, its impact on spurious bias is similar to no suppression at all. With nonzero masking values, models can still adjust their weights using biased activations, resulting in persistent spurious bias.

### 6. Conclusion

We proposed a self-guided spurious bias mitigation method that directly intervenes the prediction mechanisms within a model without using group labels. Our method exploits distinct patterns of neuron activations in a model's latent space to identify biased dimensions and suppresses signals from these dimensions while tuning the remaining model. We theoretically validated our neuron identification method and proved that our method can bring a model closer to an unbiased one than its ERM counterpart. Experiments validated our theoretical findings and showed that our method is a lightweight post hoc bias mitigation method that can work across different data modalities and model architectures. Future work may explore different choices of identification and model tuning data to enhance spurious bias mitigation.

## Acknowledgments

This work is supported in part by the US National Science Foundation under grants 2217071, 2213700, 2106913, 2008208. Any opinions, findings, and conclusions or recommendations expressed in this material are those of the author(s) and do not necessarily reflect the views of the National Science Foundation.

## Impact Statement

This paper presents work whose goal is to advance the field of Machine Learning by improving model robustness through self-guided spurious bias mitigation. Our method NeuronTune reduces reliance on spurious correlations without requiring additional annotations, thereby enhancing generalization across diverse data distributions. This has important implications for trustworthiness and reliability in AI systems, particularly in high-stakes applications where biased model predictions can lead to unintended consequences. While our work contributes positively by mitigating spurious correlations, we acknowledge that no method is entirely free from potential biases introduced during training or deployment. We encourage further research on assessing and addressing residual biases in real-world settings. However, we do not identify any specific ethical concerns that require special attention beyond standard considerations in trustworthy machine learning.

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

# A. Appendix

The appendix is organized as follows:

## A.1. Details of the Synthetic Experiment

**Data Model.** Without loss of generality, we considered an input $\mathbf{v} \in \mathbb{R}^3$ to simulate a latent embedding before the last prediction layer, which consists of three dimensions: a core dimension with the core component $v^c \in \mathbb{R}$, a spurious dimension with the spurious component $v^s \in \mathbb{R}$, and a noise dimension with the noise component $v^\epsilon$. We considered a dataset $\mathcal{D}^{\text{syn}} = \{(\mathbf{v}_i, y_i)\}_{i=1}^N$ of $N$ sample-label pairs, where $y_i \in \{-1, +1\}$, $v_i^c = y_i + n_c$, and $v^\epsilon$ and $n_c$ are zero-mean Gaussian noises with variances $\sigma_\epsilon^2$ and $\sigma_c^2$, respectively. When $y_i = -1$, $v_i^s = 0 + n_s$ with the probability $\alpha$ and $v_i^s = 1 + n_s$ with the probability $1 - \alpha$; when $y_i = +1$, $v_i^s = 1 + n_s$ with the probability $\alpha$ and $v_i^s = 0 + n_s$ with the probability $1 - \alpha$, where $n_s$ is an independent zero-mean Gaussian noise with the variance $\sigma_s^2$. To facilitate developing the spurious bias of using the correlation between $v_i^s$ and $y_i$ for predictions, we generated a training set $\mathcal{D}_{\text{train}}^{\text{syn}}$ with easy-to-learn spurious attributes by setting $\sigma_c^2 > \sigma_s^2$ and $\alpha \approx 1$ (Sagawa et al., 2020). Thus, the correlations between $v_i^s$ and $y_i$ are predictive of $\alpha N$ labels. To demonstrate, we set $\sigma_c^2 = 0.6$, $\sigma_s^2 = 0.1$, $\sigma_\epsilon^2 = 0.1$, $\alpha = 0.95$, and $N = 5000$. We generated a test set $\mathcal{D}_{\text{test}}^{\text{syn}}$ with the same set of parameters except $\alpha = 0.1$. Now, spurious correlations between $v_i^s$ and $y_i$ are only predictive of a small portion of the test samples. Fig. 2 shows four data groups along with their respective proportions in each class.

**Classification Model.** We considered a logistic regression model $\phi_{\tilde{\mathbf{w}}}(\mathbf{v}) = 1/(1 + \exp\{-(\mathbf{w}^T \mathbf{v} + b)\})$, where $\tilde{\mathbf{w}} = [\mathbf{w}, b]$. The model predicts $+1$ when $\phi_{\tilde{\mathbf{w}}}(\mathbf{v}) > 0.5$ and $-1$ otherwise. We trained $\phi_{\tilde{\mathbf{w}}}$ on $\mathcal{D}_{\text{train}}^{\text{syn}}$ and tested it on $\mathcal{D}_{\text{test}}^{\text{syn}}$.

**Spurious Bias.** We observed a high average accuracy of 95.4% but a WGA of 66.2% (Fig. 2(a) in the main paper) on the training data. The results show that the model heavily relies on the correlations that exist in the majority of samples and exhibits strong spurious bias. As expected, the performance on the test data is significantly lower (Fig. 2(a), right). The decision boundary (Fig. 2(a), black lines) learned from the training data does not generalize to the test data.

**Mitigation Strategy.** Without group labels, it is challenging to identify and mitigate spurious bias in the model. We tackled this challenge by first finding that the distributions of values of an input dimension, together with the prediction outcomes for a certain class, provide discriminative information regarding the spuriousness of the dimension. (1) When the values for misclassified samples at the dimension are high, while values for the correctly predicted samples are low, this indicates that

the absence of the dimension input does not significantly affect the correctness of predictions, while the presence of the dimension input does not generalize to certain groups of data. Therefore, the dimension tends to be a biased dimension. The plots in Fig. 2(b) illustrate the value distributions of the first and second dimensions of input embeddings when $y_i = -1$. (2) In contrast, if the absence of the dimension input results in misclassification, then the dimension tends to represent a core attribute. The left plot of Fig. 2(b) represents the first dimension of input embeddings when $y_i = -1$. Next, we retrained the model while suppressing the second and third dimensions. As a result, the retrained model has learned to balance its performance on both the training and test data with a significant increase in WGA on the test data (Fig. 2(c)).

## A.2. Theoretical Analysis

### A.2.1. PRELIMINARY

For the ease of readability, we restate the data model specified by (1) and (2) in the following

$$\mathbf{x} = \mathbf{x}_{\text{core}} \oplus \mathbf{x}_{\text{spu}} \in \mathbb{R}^{D \times 1}, \ y = \boldsymbol{\beta}^T \mathbf{x}_{\text{core}} + \varepsilon_{\text{core}}, \tag{8}$$

and

$$\mathbf{x}_{\text{spu}} = (2a - 1)\boldsymbol{\gamma}y + \boldsymbol{\varepsilon}_{\text{spu}}, a \sim \text{Bern}(p), \tag{9}$$

where $(2a - 1) \in \{-1, +1\}$, $a \sim \text{Bern}(p)$ is a Bernoulli random variable, $p$ is close to 1, $\varepsilon_{\text{core}}$ is a zero-mean Gaussian random variable with the variance $\eta^2_{\text{core}}$, and each element in $\boldsymbol{\varepsilon}_{\text{spu}}$ follows a zero-mean Gaussian distribution with the variance $\eta^2_{\text{spu}}$. We set $\eta^2_{\text{core}} \gg \eta^2_{\text{spu}}$ to facilitate the learning of spurious attributes. The model $f(\mathbf{x}) = \mathbf{b}^T \mathbf{W} \mathbf{x}$ in Section 4.1 can be further expressed as follows,

$$\hat{y} = \sum_{i=1}^{M} b_i(\mathbf{x}_{\text{core}}^T \mathbf{w}_{\text{core},i} + \mathbf{x}_{\text{spu}}^T \mathbf{w}_{\text{spu},i}) = \mathbf{x}_{\text{core}}^T \mathbf{u}_{\text{core}} + \mathbf{x}_{\text{spu}}^T \mathbf{u}_{\text{spu}}, \tag{10}$$

where $\mathbf{w}_i^T \in \mathbb{R}^{1 \times D}$ is the $i$-th row of $\mathbf{W}$, $\mathbf{w}_i^T = [\mathbf{w}_{\text{core},i}^T, \mathbf{w}_{\text{spu},i}^T]$ with $\mathbf{w}_{\text{core},i} \in \mathbb{R}^{D_1 \times 1}$ and $\mathbf{w}_{\text{spu},i} \in \mathbb{R}^{D_2 \times 1}$, $\mathbf{u}_{\text{core}} = \sum_{i=1}^{M} b_i \mathbf{w}_{\text{core},i}$, and $\mathbf{u}_{\text{spu}} = \sum_{i=1}^{M} b_i \mathbf{w}_{\text{spu},i}$. The loss function which we use to optimize $\mathbf{W}$ and $\mathbf{b}$ is

$$\ell_{\text{tr}}(\mathbf{W}, \mathbf{b}) = \frac{1}{2} \mathbb{E}_{(\mathbf{x},y) \in \mathcal{D}_{\text{train}}} \|f(\mathbf{x}) - y\|_2^2. \tag{11}$$

With the above definitions, the following lemma gives the optimal coefficients $\mathbf{u}_{\text{core}}^*$ and $\mathbf{u}_{\text{spu}}^*$ based on the training data.

### A.2.2. PROOF OF LEMMA 1

**Lemma 1.** *Given a training dataset $\mathcal{D}_{\text{train}}$ with $p$ defined in (9) satisfying $1 \geq p \gg 0.5$, the optimized weights in the form of $\mathbf{u}_{\text{core}}^*$ and $\mathbf{u}_{\text{spu}}^*$ are*

$$\mathbf{u}_{core}^* = \frac{(2 - 2p)\eta_{core}^2 + \eta_{spu}^2}{\eta_{core}^2 + \eta_{spu}^2} \boldsymbol{\beta}, \tag{12}$$

*and*

$$\mathbf{u}_{spu}^* = \frac{(2p - 1)\eta_{core}^2}{\eta_{core}^2 + \eta_{spu}^2} \boldsymbol{\gamma}, \tag{13}$$

*respectively. When $p = 0.5$, the training data is unbiased and we obtain an unbiased classifier with weights $\mathbf{u}_{core}^* = \boldsymbol{\beta}$ and $\mathbf{u}_{spu}^* = 0$.*

*Proof.* Note that $f(\mathbf{x}) = \mathbf{b}^T \mathbf{W} \mathbf{x} = \mathbf{x}^T \mathbf{v} = \mathbf{x}_{\text{core}}^T \mathbf{u}_{\text{core}} + \mathbf{x}_{\text{spu}}^T \mathbf{u}_{\text{spu}}$, then we have

$$\ell_{\text{tr}}(W, b) = \frac{1}{2} \mathbb{E} \|\mathbf{x}_{\text{core}}^T \mathbf{u}_{\text{core}} + \mathbf{x}_{\text{spu}}^T \mathbf{u}_{\text{spu}} - y\|_2^2 \tag{14}$$

$$= \frac{1}{2} \mathbb{E} \|\mathbf{x}_{\text{core}}^T \mathbf{u}_{\text{core}} + \left[(2a - 1)\boldsymbol{\gamma}y + \boldsymbol{\varepsilon}_{\text{spu}}\right]^T \mathbf{u}_{\text{spu}} - y\|_2^2 \tag{15}$$

$$= \frac{1}{2} \mathbb{E} \|\mathbf{x}_{\text{core}}^T \mathbf{u}_{\text{core}} - \left[1 - (2a - 1)\boldsymbol{\gamma}^T \mathbf{u}_{\text{spu}}\right]y\|_2^2 + \frac{1}{2}\eta_{\text{spu}}^2 \|\mathbf{u}_{\text{spu}}\|_2^2 \tag{16}$$

$$= \frac{1}{2}(pE_1 + (1 - p)E_2) + \frac{1}{2}\eta_{\text{spu}}^2 \|\mathbf{u}_{\text{spu}}\|_2^2, \tag{17}$$

where $E_1 = \|\mathbf{x}_{\text{core}}^T \mathbf{u}_{\text{core}} - (1 - \boldsymbol{\gamma}^T \mathbf{u}_{\text{spu}})y\|_2^2$ when $a = 1$ and $E_2 = \|\mathbf{x}_{\text{core}}^T \mathbf{u}_{\text{core}} - (1 + \boldsymbol{\gamma}^T \mathbf{u}_{\text{spu}})y\|_2^2$ when $a = 0$. We first calculate the lower bound for $E_1$ as follows

$$E_1 = \mathbb{E}\|\mathbf{x}_{\text{core}}^T \mathbf{u}_{\text{core}} - (1 - \boldsymbol{\gamma}^T \mathbf{u}_{\text{spu}})(\boldsymbol{\beta}^T \mathbf{x}_{\text{core}} + \varepsilon_{\text{core}})\|_2^2 \tag{18}$$

$$= \mathbb{E}\|\mathbf{x}_{\text{core}}^T \mathbf{u}_{\text{core}} - (1 - \boldsymbol{\gamma}^T \mathbf{u}_{\text{spu}})\boldsymbol{\beta}^T \mathbf{x}_{\text{core}} + (1 - \boldsymbol{\gamma}^T \mathbf{u}_{\text{spu}})\varepsilon_{\text{core}})\|_2^2 \tag{19}$$

$$= \mathbb{E}\|\mathbf{x}_{\text{core}}^T \mathbf{u}_{\text{core}} - (1 - \boldsymbol{\gamma}^T \mathbf{u}_{\text{spu}})\boldsymbol{\beta}^T \mathbf{x}_{\text{core}}\|_2^2 + \eta_{\text{core}}^2(1 - \boldsymbol{\gamma}^T \mathbf{u}_{\text{spu}})^2 \tag{20}$$

$$\geq \eta_{\text{core}}^2(1 - \boldsymbol{\gamma}^T \mathbf{u}_{\text{spu}})^2. \tag{21}$$

Similarly, we have

$$E_2 = \mathbb{E}\|\mathbf{x}_{\text{core}}^T \mathbf{u}_{\text{core}} - (1 + \boldsymbol{\gamma}^T \mathbf{u}_{\text{spu}})(\boldsymbol{\beta}^T \mathbf{x}_{\text{core}} + \varepsilon_{\text{core}})\|_2^2 \tag{22}$$

$$= \mathbb{E}\|\mathbf{x}_{\text{core}}^T \mathbf{u}_{\text{core}} - (1 + \boldsymbol{\gamma}^T \mathbf{u}_{\text{spu}})\boldsymbol{\beta}^T \mathbf{x}_{\text{core}}\|_2^2 + \eta_{\text{core}}^2(1 + \boldsymbol{\gamma}^T \mathbf{u}_{\text{spu}})^2 \tag{23}$$

$$\geq \eta_{\text{core}}^2(1 + \boldsymbol{\gamma}^T \mathbf{u}_{\text{spu}})^2. \tag{24}$$

Then, plug in (21) and (24) into (17), we obtain the following

$$\ell_{\text{tr}}(W, b) \geq \frac{1}{2}\Big(p\eta_{\text{core}}^2(1 - \boldsymbol{\gamma}^T \mathbf{u}_{\text{spu}})^2 + (1 - p)\eta_{\text{core}}^2(1 + \boldsymbol{\gamma}^T \mathbf{u}_{\text{spu}})^2 + \eta_{\text{spu}}^2\|\mathbf{u}_{\text{spu}}\|_2^2\Big) \tag{25}$$

$$= \frac{1}{2}\Big(p\eta_{\text{core}}^2(1 - \boldsymbol{\gamma}^T \mathbf{u}_{\text{spu}})^2 + (1 - p)\eta_{\text{core}}^2(1 + \boldsymbol{\gamma}^T \mathbf{u}_{\text{spu}})^2 + \eta_{\text{spu}}^2\|\boldsymbol{\gamma}\|_2^2\|\mathbf{u}_{\text{spu}}\|_2^2\Big) \tag{26}$$

$$\geq \frac{1}{2}\Big(p\eta_{\text{core}}^2(1 - \boldsymbol{\gamma}^T \mathbf{u}_{\text{spu}})^2 + (1 - p)\eta_{\text{core}}^2(1 + \boldsymbol{\gamma}^T \mathbf{u}_{\text{spu}})^2 + \eta_{\text{spu}}^2\|\boldsymbol{\gamma}^T \mathbf{u}_{\text{spu}}\|_2^2\Big), \tag{27}$$

where (26) uses the fact that $\boldsymbol{\gamma}$ has a unit norm, and the inequality (27) exploits the Cauchy–Schwarz inequality. Let $z = \boldsymbol{\gamma}^T \mathbf{u}_{\text{spu}}$, we have $\ell(z) = p\eta_{\text{core}}^2(1 - z)^2 + (1 - p)\eta_{\text{core}}^2(1 + z)^2 + \eta_{\text{spu}}^2 z^2$. Let $\frac{\partial \ell(z)}{\partial z} = 0$, we obtain

$$z^* = \boldsymbol{\gamma}^T \mathbf{u}_{\text{spu}}^* = \frac{(2p - 1)\eta_{\text{core}}^2}{\eta_{\text{core}}^2 + \eta_{\text{spu}}^2}.$$

Given $\mathbf{u}_{\text{spu}}^*$, we can obtain the optimal $\mathbf{u}_{\text{core}}'$ for minimizing $E_1$ in (20) as $\mathbf{u}_{\text{core}}' = (1 - z^*)\boldsymbol{\beta}$; similarly, we can obtain the optimal $\mathbf{u}_{\text{core}}''$ for minimizing $E_2$ in (23) as $\mathbf{u}_{\text{core}}'' = (1 + z^*)\boldsymbol{\beta}$. Via proof by contradiction, only $\mathbf{u}_{\text{core}}'$ or $\mathbf{u}_{\text{core}}''$ is the solution for $\mathbf{u}_{\text{core}}^*$. Since $p \gg 0.5$, $E_1$ contributes to the majority error of (20). Thus, $\mathbf{u}_{\text{core}}^* = (1 - z^*)\boldsymbol{\beta}$, i.e.,

$$\mathbf{u}_{\text{core}}^* = (1 - z^*)\boldsymbol{\beta} = \frac{(2 - 2p)\eta_{\text{core}}^2 + \eta_{\text{spu}}^2}{\eta_{\text{core}}^2 + \eta_{\text{spu}}^2}\boldsymbol{\beta}.$$

$\square$

### A.2.3. PROOF OF COROLLARY 1

Lemma 1 gives the optimal model weights under a given training dataset $\mathcal{D}_{\text{train}}$ with the parameter $p$ controlling the strength of spurious correlations. Lemma 1 generalizes the result in (Ye et al., 2023) where $p = 1$. Importantly, we obtain the following corollary for unbiased models:

**Corollary 1.** *The unbiased model $f(\mathbf{x}) = \mathbf{u}^T \mathbf{x} = \mathbf{x}_{core}^T \mathbf{u}_{core} + \mathbf{x}_{spu}^T \mathbf{u}_{spu}$ is achieved when $\mathbf{u}_{core} = \mathbf{u}_{core}^*$ and $\boldsymbol{\gamma}^T \mathbf{u}_{spu} = 0$.*

*Proof.* Plug $\boldsymbol{\gamma}^T \mathbf{u}_{\text{core}} = 0$ into (20) and (23), then we observe that $\mathbf{u}_{\text{core}}$ minimizes errors from both the majority ($a = 1$) and minority ($a = 0$) groups of data. $\square$

If we could obtain a set of unbiased training data with $p = 0.5$, then we obtain an unbiased model with $\mathbf{u}_{\text{spu}}^* = 0$ and $\mathbf{u}_{\text{core}}^* = \boldsymbol{\beta}$. However, in practice, it is challenging to obtain a set of unbiased training data, i.e., it is challenging to control the value of $p$.

### A.2.4. PROOF OF PROPOSITION 4.1

**Proposition 4.1** (**Principle of NeuronTune**). *Given the model $f(\mathbf{x}) = \mathbf{b}^T \mathbf{W} \mathbf{x}$ trained with data generated under the data model specified in* (8) *and* (9), *it captures spurious correlations when $\boldsymbol{\gamma}^T \mathbf{w}_{spu,i} < 0, i \in \{1, \ldots, M\}$. The principle of NeuronTune is to suppress neurons containing negative $\boldsymbol{\gamma}^T \mathbf{w}_{spu,i}$.*

*Proof.* Consider the $i$-th neuron $e_i$ $(i = 1, \ldots, M)$ before the last layer. We first expand it based on our data model specified by (8) and (9) as follows:

$$e_i = \mathbf{x}_{\text{core}}^T \mathbf{w}_{\text{core},i} + \mathbf{x}_{\text{spu}}^T \mathbf{w}_{\text{spu},i} \tag{28}$$

$$= \mathbf{x}_{\text{core}}^T \mathbf{w}_{\text{core},i} + [(2a-1)\boldsymbol{\gamma}y + \boldsymbol{\varepsilon}_{\text{spu}}]^T \mathbf{w}_{\text{spu},i} \tag{29}$$

$$= \mathbf{x}_{\text{core}}^T \mathbf{w}_{\text{core},i} + (2a-1)[\boldsymbol{\beta}^T \mathbf{x}_{\text{core}} + \varepsilon_{\text{core}}]\boldsymbol{\gamma}^T \mathbf{w}_{\text{spu},i} + \boldsymbol{\varepsilon}_{\text{spu}}^T \mathbf{w}_{\text{spu},i} \tag{30}$$

$$= \mathbf{x}_{\text{core}}^T \mathbf{w}_{\text{core},i} + (2a-1)\boldsymbol{\beta}^T \mathbf{x}_{\text{core}}\boldsymbol{\gamma}^T \mathbf{w}_{\text{spu},i} + \varepsilon_{\text{rem}}, \tag{31}$$

where $\varepsilon_{\text{rem}} = \varepsilon_{\text{core}}\boldsymbol{\gamma}^T \mathbf{w}_{\text{spu},i} + \boldsymbol{\varepsilon}_{\text{spu}}^T \mathbf{w}_{\text{spu},i}$. In (31), if $\boldsymbol{\gamma}^T \mathbf{w}_{\text{spu},i} \geq 0$, the model handles the spurious component correctly. Specifically, when $a = 1$, the spurious component positively correlates with the core component and contributes to the output, whereas when $a = 0$, its correlation with the core component breaks with a negative one and has a negative contribution to the output. In contrast, if $\boldsymbol{\gamma}^T \mathbf{w}_{\text{spu},i} < 0$ and $a = 1$, then the model still utilizes the spurious component even the correlation breaks, demonstrating a strong reliance on the spurious component instead of the core component. Therefore, the principle of selective activation is to find neurons containing negative $\boldsymbol{\gamma}^T \mathbf{w}_{\text{spu},i}$ so that suppressing them improves model generalization. $\square$

### A.2.5. PROOF OF THEOREM 4.2

The following theorem validates our neuron selection method.

**Theorem 4.2** (**Metric for Neuron Selection**). *Given the model $f(\mathbf{x}) = \mathbf{b}^T \mathbf{W} \mathbf{x}$, we cast it to a classification model by training it to regress $y \in \{-\mu, \mu\}$ $(\mu > 0)$ on $\mathbf{x}$ based on the data model specified in* (8) *and* (9)*, where $\mu = \mathbb{E}[\boldsymbol{\beta}^T \mathbf{x}_{core}]$. The metric $\delta_i^y$ defined in the following can identify neurons with spurious correlations when $\delta_i^y > 0$:*

$$\delta_i^y = Med(\bar{\mathcal{V}}_i^y) - Med(\hat{\mathcal{V}}_i^y),$$

*where $\bar{\mathcal{V}}_i^y$ and $\hat{\mathcal{V}}_i^y$ are the sets of activation values for misclassified and correctly predicted samples with the label $y$ from the $i$-th neuron, respectively; an activation value is defined as $\mathbf{x}_{core}^T \mathbf{w}_{core,i} + \mathbf{x}_{spu}^T \mathbf{w}_{spu,i}$; and $Med(\cdot)$ returns the median of an input set of values.*

*Proof.* We start by obtaining the set of correctly predicted samples $\hat{\mathcal{D}}_y$ and the set of incorrectly predicted samples $\bar{\mathcal{D}}_y$ as $\hat{\mathcal{D}}_y = \{\mathbf{x}|f(\mathbf{x}) \geq 0, (\mathbf{x}, y) \in \mathcal{D}_{\text{Ide}}\}$ and $\bar{\mathcal{D}}_y = \{\mathbf{x}|f(\mathbf{x}) < 0, (\mathbf{x}, y) \in \mathcal{D}_{\text{Ide}}\}$, where $\mathcal{D}_{\text{Ide}}$ is the set of identification data. Then, we have $\hat{\mathcal{V}}_i^y = \{e_i|\mathbf{x} \in \hat{\mathcal{D}}_y\}$, and $\bar{\mathcal{V}}_i^y = \{e_i|\mathbf{x} \in \bar{\mathcal{D}}_y\}$, where $e_i$ is the $i$-th neuron activation defined in (31). Expanding $e_i$ following (31), we obtain

$$e_i = \mathbf{x}_{\text{core}}^T \mathbf{w}_{\text{core},i} + (2a-1)\boldsymbol{\beta}^T \mathbf{x}_{\text{core}}\boldsymbol{\gamma}^T \mathbf{w}_{\text{spu},i} + \varepsilon_{\text{rem}}.$$

Note that $\mathbf{x}_{\text{core}}^T \mathbf{w}_{\text{core},i}$ and $\varepsilon_{\text{rem}}$ exist for all the samples, regardless of the ultimate prediction results, and all $e_i$ follows a Gaussian distribution given $a$. Then, among all the correctly predicted samples with the label $y$, according the Lemma 2, we have $\text{Med}(\hat{\mathcal{V}}_i^y) \approx \mathbb{E}[\mathbf{x}_{\text{core}}^T \mathbf{w}_{\text{core},i}] + \mu\boldsymbol{\gamma}^T \mathbf{w}_{\text{spu},i}$. Similarly, among all the incorrectly predicted samples with the label $y$, we have $\text{Med}(\bar{\mathcal{V}}_i^y) \approx \mathbb{E}[\mathbf{x}_{\text{core}}^T \mathbf{w}_{\text{core},i}] - \mu\boldsymbol{\gamma}^T \mathbf{w}_{\text{spu},i}$. Then, the difference between the two is

$$\delta_i^y \approx -2\mu\boldsymbol{\gamma}^T \mathbf{w}_{\text{spu},i}.$$

When $\delta_i^y > 0$, we have $\boldsymbol{\gamma}^T \mathbf{w}_{\text{spu},i} < 0$. According Proposition 4.1, using $\delta_i^y > 0$ indeed selects neurons that have strong reliance on spurious components. $\square$

A.2.6. PROOF OF THEOREM 4.3

**Theorem 4.3** (**NeuronTune Mitigates Spurious Bias**). *Consider the model $f^*(\mathbf{x}) = \mathbf{x}^T \mathbf{u}^*$ trained on the biased training data with $p \gg 0.5$, with $\mathbf{u}^*_{core}$ and $\mathbf{u}^*_{spu}$ defined in (12) and (13), respectively. Under the mild assumption that $\boldsymbol{\beta}^T \mathbf{w}_{core,i} \approx \boldsymbol{\gamma}^T \mathbf{w}_{spu,i}, \forall i = 1, \ldots, M$, then applying NeuronTune to $f^*(\mathbf{x})$ produces a model that is closer to the unbiased one.*

*Proof.* Consider $f^*(\mathbf{x})$ as the base model. We aim to prove that the retrained model obtained with NeuronTune is closer to the unbiased model defined in Corollary 1 than the base model in the parameter space.

First, the assumption that $\boldsymbol{\beta}^T \mathbf{w}_{core,i} \approx \boldsymbol{\gamma}^T \mathbf{w}_{spu,i}, \forall i = 1, \ldots, M$ generally holds for a biased model as the model has learned to associate spurious attributes with the core attributes.

Then, we denote the retrained parameters obtained with NeuronTune as $\mathbf{u}^\dagger_{core}$ and $\mathbf{u}^\dagger_{spu}$. We start with calculating $\mathbf{u}^\dagger_{spu}$. Focusing on (27) and following the derivation in Lemma 1, we obtain $\mathbf{u}^\dagger_{spu} = \sum_{i \in \mathcal{I}_+} b_i \mathbf{w}_{spu,i} = \mathbf{u}^*_{spu}$, where $\mathcal{I}_+$ denotes the set of neuron indexes satisfying $\boldsymbol{\gamma}^T \mathbf{w}_{spu,i} > 0$. Note that NeuronTune is a last-layer retraining method; thus we only optimize $b_i$ here and $\mathbf{w}_{spu,i}$ is the same as in $f^*(\mathbf{x})$. Left multiplying $\mathbf{u}^\dagger_{spu}$ with $\boldsymbol{\gamma}^T$, we have

$$\boldsymbol{\gamma}^T \mathbf{u}^\dagger_{spu} = \sum_{i \in \mathcal{I}_+} b_i^\dagger \boldsymbol{\gamma}^T \mathbf{w}_{spu,i} \tag{32}$$

$$= z^* = \frac{(2p-1)\eta^2_{core}}{\eta^2_{core} + \eta^2_{spu}} > 0.$$

Note that $\boldsymbol{\gamma}^T \mathbf{w}_{spu,i} > 0, \forall i \in \mathcal{I}_+$ because of NeuronTune. Hence, we have $b_i^\dagger > 0, \forall i \in \mathcal{I}_+$. Moreover, we observe that $\mathbf{u}^\dagger_{spu}$ is the same as $\mathbf{u}^*_{spu}$ as long as $\mathcal{I}_+$ is non-empty. This shows that NeuronTune is not able to optimize parameters related to the spurious components in the input data.

According to the Corollary 1, the unbiased model is achieved when $p = 0.5$ and $\mathbf{u}_{core} = \boldsymbol{\beta}$. The Euclidean distance between $\boldsymbol{\beta}$ and the biased solution $\mathbf{u}_{core} = (1 - z^*)\boldsymbol{\beta}$ is $\|\mathbf{u}^*_{core} - \boldsymbol{\beta}\| = z^*$. Based on (32), we estimate the distance between our NeuronTune solution $\mathbf{u}^\dagger_{core}$ and $\boldsymbol{\beta}$ as follows

$$\|\mathbf{u}^\dagger_{core} - \boldsymbol{\beta}\|_2 = \|\boldsymbol{\beta}^T(\mathbf{u}^\dagger_{core} - \boldsymbol{\beta})\|_2 \tag{33}$$

$$= \|\boldsymbol{\beta}^T \mathbf{u}^\dagger_{core} - 1\|_2 \tag{34}$$

$$= \|\sum_{i \in \mathcal{I}_+} b_i^\dagger \boldsymbol{\beta}^T \mathbf{w}_{core,i} - 1\|_2 \tag{35}$$

$$\approx \|\sum_{i \in \mathcal{I}_+} b_i^\dagger \boldsymbol{\gamma}^T \mathbf{w}_{spu,i} - 1\|_2 \tag{36}$$

$$= \|z^* - 1\|, \tag{37}$$

where (34) uses the fact that $\boldsymbol{\beta}^T \boldsymbol{\beta} = 1$, and (35) uses the condition $\boldsymbol{\beta}^T \mathbf{w}_{core,i} \approx \boldsymbol{\gamma}^T \mathbf{w}_{spu,i}, \forall i = 1, \ldots, M$. Note that $z^*$ is achieved on the training data with $p \gg 0.5$ and $\eta^2_{core} \gg \eta^2_{spu}$, hence we have $z^* \approx 1$ and $\|\mathbf{u}^\dagger_{core} - \boldsymbol{\beta}\|_2 \approx 0$. In other words, NeuronTune can bring model parameters closer to the optimal and unbiased solution than the parameters of the biased model.

□

A.2.7. PROOF OF LEMMA 2

**Lemma 2** (**Majority of Samples among Different Predictions**). *Given the model $f(\mathbf{x}) = \mathbf{b}^T \mathbf{W} \mathbf{x}$ trained on $y \in \{-\mu, \mu\}$ ($\mu > 0$) with $\mu = \mathbb{E}[\boldsymbol{\beta}^T \mathbf{x}_{core}]$, and the conditions that $p > 3/4$ and $\eta^2_{core} \gg \eta^2_{spu}$, we have the following claims:*

- *Among the set of all correctly predicted samples with the label $y$, more than half of them are generated with $a = 1$;*

- *Among the set of all incorrectly predicted samples with the label $y$, more than half of them are generated with $a = 0$.*

*Proof.* With the two regression targets, $-\mu$ and $\mu$, the optimal decision boundary is 0. Without loss of generality, we consider $y = \mu$. Then, the set of correctly predicted samples $\hat{\mathcal{D}}_y$ is

$$\hat{\mathcal{D}}_y = \{\mathbf{x}|f(\mathbf{x}) \geq 0, (\mathbf{x}, y) \in \mathcal{D}_{\text{Ide}}\},$$

and the set of incorrectly predicted samples $\hat{\mathcal{D}}_y$ is

$$\bar{\mathcal{D}}_y = \{\mathbf{x}|f(\mathbf{x}) < 0, (\mathbf{x}, y) \in \mathcal{D}_{\text{Ide}}\}.$$

The probability of a sample with the label $y$ that is correctly predicted is

$$P(\mathbf{x} \in \hat{\mathcal{D}}_y|y) = P(a = 1)P(f(\mathbf{x}) \geq 0|a = 1, y) + P(a = 0)P(f(\mathbf{x}) \geq 0|a = 0, y)$$
$$= pP(f(\mathbf{x}) \geq 0|a = 1, y) + (1 - p)P(f(\mathbf{x}) \geq 0|a = 0, y).$$

Similarly, the probability of a sample with the label $y$ that is incorrectly predicted is

$$P(\mathbf{x} \in \bar{\mathcal{D}}_y|y) = pP(f(\mathbf{x}) < 0|a = 1, y) + (1 - p)P(f(\mathbf{x}) < 0|a = 0, y).$$

To calculate $P(f(\mathbf{x}) \geq 0|a = 1, y)$, we expand $f(\mathbf{x})$ as follows:

$$f(\mathbf{x}) = \mathbf{x}_{\text{core}}^T \mathbf{u}_{\text{core}}^* + \mathbf{x}_{\text{spu}}^T \mathbf{u}_{\text{spu}}^*$$
$$= \mathbf{x}_{\text{core}}^T \boldsymbol{\beta}(1 - z^*) + (\boldsymbol{\gamma}(\boldsymbol{\beta}^T \mathbf{x}_{\text{core}} + \varepsilon_{\text{core}}) + \boldsymbol{\varepsilon}_{\text{spu}})^T \mathbf{u}_{\text{spu}}^*$$
$$= \mathbf{x}_{\text{core}}^T \boldsymbol{\beta}(1 - z^*) + \mathbf{x}_{\text{core}}^T \boldsymbol{\beta}\boldsymbol{\gamma}^T \mathbf{u}_{\text{spu}}^* + \boldsymbol{\gamma}^T \mathbf{u}_{\text{spu}}^* \varepsilon_{\text{core}} + \boldsymbol{\varepsilon}_{\text{spu}}^T \mathbf{u}_{\text{spu}}^*$$
$$= \mathbf{x}_{\text{core}}^T \boldsymbol{\beta} + z^* \varepsilon_{\text{core}} + \boldsymbol{\varepsilon}_{\text{spu}}^T \mathbf{u}_{\text{spu}}^*.$$

The output of $f(\mathbf{x})$ follows a Gaussian distribution, with the mean $\mu_1 = \mathbb{E}[f(\mathbf{x})] = \mu$, and the variance $\sigma_1^2 = Var(\mathbf{x}_{\text{core}}^T \boldsymbol{\beta}) + \eta_{\text{core}}^2(z^*)^2 + \eta_{\text{spu}}^2(z^*)^2$. Therefore, we have

$$P(f(\mathbf{x}) \geq 0|a = 1, y) = P(\mathbf{x} \in \hat{\mathcal{D}}_y|a = 1, y) = 1 - \Phi(\frac{0 - \mu}{\sigma_1}) = \Phi(\frac{\mu}{\sigma_1}), \tag{38}$$

$$P(f(\mathbf{x}) < 0|a = 1, y) = P(\mathbf{x} \in \bar{\mathcal{D}}_y|a = 1, y) = 1 - \Phi(\frac{\mu}{\sigma_1}) = \Phi(\frac{-\mu}{\sigma_1}). \tag{39}$$

Similarly, to calculate $P(f(\mathbf{x}) \geq 0|a = 0, y)$, we expand $f(\mathbf{x})$ as follows:

$$f(\mathbf{x}) = \mathbf{x}_{\text{core}}^T \boldsymbol{\beta}(1 - z^*) - \mathbf{x}_{\text{core}}^T \boldsymbol{\beta}\boldsymbol{\gamma}^T \mathbf{u}_{\text{spu}}^* - \boldsymbol{\gamma}^T \mathbf{u}_{\text{spu}}^* \varepsilon_{\text{core}} + \boldsymbol{\varepsilon}_{\text{spu}}^T \mathbf{u}_{\text{spu}}^*$$
$$= \mathbf{x}_{\text{core}}^T \boldsymbol{\beta}(1 - 2z^*) - z^* \varepsilon_{\text{core}} + \boldsymbol{\varepsilon}_{\text{spu}}^T \mathbf{u}_{\text{spu}}^*.$$

The output of $f(\mathbf{x})$ follows a Gaussian distribution, with the mean $\mu_0 = \mathbb{E}[f(\mathbf{x})] = \mu(1 - 2z^*)$, and the variance $\sigma_0^2 = (1 - 2z^*)^2 Var(\mathbf{x}_{\text{core}}^T \boldsymbol{\beta}) + \eta_{\text{core}}^2(z^*)^2 + \eta_{\text{spu}}^2(z^*)^2$. As a result, we have

$$P(f(\mathbf{x}) \geq 0|a = 0, y) = P(x \in \hat{\mathcal{D}}_y|a = 0, y) = 1 - \Phi(\frac{0 - \mu_0}{\sigma_0}) = \Phi(\frac{(1 - 2z^*)\mu}{\sigma_0}), \tag{40}$$

$$P(f(\mathbf{x}) < 0|a = 0, y) = P(x \in \bar{\mathcal{D}}_y|a = 0, y) = 1 - \Phi(\frac{\mu_0}{\sigma_0}) = \Phi(\frac{-(1 - 2z^*)\mu}{\sigma_0}). \tag{41}$$

Then, we obtain the probabilities for correctly and incorrectly predicted samples with the label $y$, i.e.,

$$P(\mathbf{x} \in \hat{\mathcal{D}}_y|y) = p\Phi(\frac{\mu}{\sigma_1}) + (1 - p)\Phi(\frac{(1 - 2z^*)\mu}{\sigma_0}), \tag{42}$$

and

$$P(\mathbf{x} \in \bar{\mathcal{D}}_y|y) = p\Phi(\frac{-\mu}{\sigma_1}) + (1 - p)\Phi(\frac{-(1 - 2z^*)\mu}{\sigma_0}). \tag{43}$$

Next, we seek to determine whether the majority of samples in the correctly (incorrectly) predicted set $\hat{\mathcal{D}}_y$ ($\bar{\mathcal{D}}_y$) is generated with $a = 0$ or $a = 1$. To achieve this, in the set of correctly predicted samples, we use the Bayesian theorem based on (42), i.e.,

$$
\begin{aligned}
P(a = 1|\mathbf{x} \in \hat{\mathcal{D}}_y, y) &= \frac{P(\mathbf{x} \in \hat{\mathcal{D}}_y|a = 1, y)P(a = 1)}{P(\mathbf{x} \in \hat{\mathcal{D}}_y|y)} \\
&= \frac{p\Phi(\mu/\sigma_1)}{p\Phi(\mu/\sigma_1) + (1 - p)\Phi((1 - 2z^*)\mu/\sigma_0)},
\end{aligned}
\tag{44}
$$

and

$$
\begin{aligned}
P(a = 0|\mathbf{x} \in \hat{\mathcal{D}}_y, y) &= 1 - P(a = 1|\mathbf{x} \in \hat{\mathcal{D}}_y, y) \\
&= \frac{(1 - p)\Phi((1 - 2z^*)\mu/\sigma_0)}{p\Phi(\mu/\sigma_1) + (1 - p)\Phi((1 - 2z^*)\mu/\sigma_0)}.
\end{aligned}
\tag{45}
$$

Similarly, in the set of incorrectly predicted samples, we have

$$
\begin{aligned}
P(a = 1|\mathbf{x} \in \bar{\mathcal{D}}_y, y) &= \frac{P(\mathbf{x} \in \bar{\mathcal{D}}_y|a = 1, y)P(a = 1)}{P(\mathbf{x} \in \bar{\mathcal{D}}_y|y)} \\
&= \frac{p\Phi(-\mu/\sigma_1)}{p\Phi(-\mu/\sigma_1) + (1 - p)\Phi(-(1 - 2z^*)\mu/\sigma_0)},
\end{aligned}
\tag{46}
$$

and

$$
\begin{aligned}
P(a = 0|\mathbf{x} \in \bar{\mathcal{D}}_y, y) &= 1 - P(a = 1|\mathbf{x} \in \bar{\mathcal{D}}_y, y) \\
&= \frac{(1 - p)\Phi(-(1 - 2z^*)\mu/\sigma_0)}{p\Phi(-\mu/\sigma_1) + (1 - p)\Phi(-(1 - 2z^*)\mu/\sigma_0)}.
\end{aligned}
\tag{47}
$$

Under the assumption that $p > 3/4$ and $\eta_{\text{core}}^2 \gg \eta_{\text{spu}}^2$, we have $1 - 2z^* = \left((3 - 4p)\eta_{\text{core}}^2 + \eta_{\text{spu}}^2\right)/(\eta_{\text{core}}^2 + \eta_{\text{spu}}^2) < 0$. Hence, $\Phi(-(1 - 2z^*)\mu/\sigma_0) < 1/2$ and $P(a = 1|\mathbf{x} \in \hat{\mathcal{D}}_y, y) > 1/2$; in other words, **among the set of all correctly predicted samples with the label $y$, more than half of them are generated with $a = 1$.**

Moreover, under the assumption that $\Phi(-\mu/\sigma_1) \approx 0$, i.e., predictions of the model have a high signal-to-noise ratio, then $P(a = 0|\mathbf{x} \in \bar{\mathcal{D}}_y, y) > 1/2$, i.e., **among the set of all incorrectly predicted samples with the label $y$, more than half of them are generated with $a = 0$.** This assumption is generally true, as $\sigma_1^2 = Var(\mathbf{x}_{\text{core}}^T\boldsymbol{\beta}) + \eta_{\text{core}}^2(z^*)^2 + \eta_{\text{spu}}^2(z^*)^2$ is typically very small when $z^*$ approaches zero given $p > 3/4$ and $\eta_{\text{core}}^2 \gg \eta_{\text{spu}}^2$. $\qquad\square$

### A.2.8. PROOF OF LEMMA 3

**Lemma 3.** *Consider the model $f(\mathbf{x}) = \mathbf{x}^T\mathbf{u}$ with $\mathbf{u} = [\mathbf{u}_{core}, \mathbf{u}_{spu}]$, the optimal solution for $\mathbf{u}_{spu}$, denoted as $\mathbf{u}_{spu}^r$, can be achieved by last-layer retraining on the retraining data with $p_{re}$ and is calculated as*

$$
\mathbf{u}_{spu}^r = \frac{(2p_{re} - 1)\eta_{core}^2}{\eta_{core}^2 + \eta_{spu}^2}\boldsymbol{\gamma}.
\tag{48}
$$

*Proof.* First, we have $f(\mathbf{x}) = \mathbf{x}^T\mathbf{u} = \mathbf{b}^T\mathbf{W}\mathbf{x}$. For last-layer retraining, $\mathbf{b}$ is optimized. Following the derivation in Lemma 1, we similarly obtain the inequality in (27) with $p = p_{\text{re}}$, i.e.,

$$
\ell(\mathbf{b}) \geq \frac{1}{2}\left(p_{\text{re}}\eta_{\text{core}}^2(1 - \boldsymbol{\gamma}^T\mathbf{u}_{\text{spu}})^2 + (1 - p_{\text{re}})\eta_{\text{core}}^2(1 + \boldsymbol{\gamma}^T\mathbf{u}_{\text{spu}})^2 + \eta_{\text{spu}}^2\|\boldsymbol{\gamma}^T\mathbf{u}_{\text{spu}}\|_2^2\right).
\tag{49}
$$

Note that the terms on the right side of the inequality are independent of any manipulation of the retraining data, such as reweighting. Then, taking the derivative of the sum of these terms with respect to $\mathbf{b}$, we obtain the following equation

$$
\boldsymbol{\gamma}^T\mathbf{W}_{\text{spu}}\mathbf{b} = \frac{(2p_{\text{re}} - 1)\eta_{\text{core}}^2}{\eta_{\text{core}}^2 + \eta_{\text{spu}}^2},
\tag{50}
$$

where $\mathbf{u}_{\text{spu}} = \mathbf{W}_{\text{spu}}\mathbf{b}$. Since $\boldsymbol{\gamma}^T\boldsymbol{\gamma} = 1$, then we have $\mathbf{u}_{\text{spu}} = \mathbf{u}_{\text{spu}}^r$. We finally verify that $\mathbf{u}_{\text{spu}}^r$ indeed minimizes the sum of the terms on the right hand side of (49). If $p_{\text{re}}$ equals to $p$ for the training data, then $\mathbf{u}_{\text{spu}}^r = \mathbf{u}_{\text{spu}}^*$ defined in (13). $\qquad\square$

## A.3. Connection to Last-Layer Retraining Methods

Although our method shares a similar setting to last-layer retraining methods, such as AFR (Qiu et al., 2023) and DFR (Kirichenko et al., 2023), our method is fundamentally different from these methods in how spurious bias is mitigated. Take AFR for an example. It, in essence, is a sample-level method and adjusts the weights of the last layer indirectly via retraining on samples with loss-related weights. Our method directly forces the weights identified as affected by spurious bias to zero, while adjusting the remaining weights with retraining.

The advantage of NeuronTune can be explained more formally in our analytic framework. First, considering the training loss in (17), we can express it as the sum of the following terms for brevity,

$$\ell_{tr}(\mathbf{W}, \mathbf{b}) = \frac{1}{2}p\mathbb{E}[\psi_1(\mathbf{u}_{\text{core}}, \mathbf{u}_{\text{spu}})] + \frac{1}{2}(1-p)\mathbb{E}[\psi_2(\mathbf{u}_{\text{core}}, \mathbf{u}_{\text{spu}})] + \frac{1}{2}\psi_3(\mathbf{u}_{\text{spu}}), \tag{51}$$

where $p$ is the data generation parameter and is a constant, and $\psi_1$, $\psi_2$, and $\psi_3$ are defined as

$$\psi_1(\mathbf{u}_{\text{core}}, \mathbf{u}_{\text{spu}}) = \mathbb{E}\|\mathbf{x}_{\text{core}}^T\mathbf{u}_{\text{core}} - (1 - \boldsymbol{\gamma}^T\mathbf{u}_{\text{spu}})\boldsymbol{\beta}^T\mathbf{x}_{\text{core}}\|_2^2,$$

$$\psi_2(\mathbf{u}_{\text{core}}, \mathbf{u}_{\text{spu}}) = \mathbb{E}\|\mathbf{x}_{\text{core}}^T\mathbf{u}_{\text{core}} - (1 + \boldsymbol{\gamma}^T\mathbf{u}_{\text{spu}})\boldsymbol{\beta}^T\mathbf{x}_{\text{core}}\|_2^2,$$

and

$$\psi_3(\mathbf{u}_{\text{spu}}) = p\eta_{\text{core}}^2(1 - \boldsymbol{\gamma}^T\mathbf{u}_{\text{spu}})^2 + (1-p)\eta_{\text{core}}^2(1 + \boldsymbol{\gamma}^T\mathbf{u}_{\text{spu}})^2 + \eta_{\text{spu}}^2\|\boldsymbol{\gamma}^T\mathbf{u}_{\text{spu}}\|_2^2,$$

respectively. Based on Lemma 3, for last-layer retraining methods in general, the optimal solution for $\mathbf{u}_{\text{spu}}$ is $\mathbf{u}_{\text{spu}}^*$, given that the retraining data follows the same distribution as the training data.

AFR changes the distribution within the first two expectation terms $\psi_1(\mathbf{u}_{\text{core}}, \mathbf{u}_{\text{spu}})$ and $\psi_2(\mathbf{u}_{\text{core}}, \mathbf{u}_{\text{spu}})$ and jointly updates $\mathbf{u}_{\text{core}}$ and $\mathbf{u}_{\text{spu}}$, while there is no optimality guarantee for $\mathbf{u}_{\text{spu}}$ ($\psi_3(\mathbf{u}_{\text{spu}})$ is not considered in AFR). By contrast, according to Theorem 4.3, NeuronTune first ensures that $\mathbf{u}_{\text{spu}}$ is optimal, then it moves $\mathbf{u}_{\text{core}}$ close the the unbiased solution.

## A.4. Comparison between Models Selected with Worst-Class Accuracy

We compared our approach with AFR (Qiu et al., 2023) and JTT (Liu et al., 2021) to demonstrate the challenges of the unsupervised setting for semi-supervised methods. These methods were tuned using worst-class accuracy (Yang et al., 2023) on the validation set instead of WGA. As shown in Table 6, our method exhibits larger performance gains over AFR and JTT compared to their results presented in Tables 1 and 2.

Table 6. WGA comparison when models selected by the worst-class accuracy on the validation set.

| Method | Waterbirds | CelebA |
|---|---|---|
| JTT | $84.2_{\pm 0.5}$ | $52.3_{\pm 1.8}$ |
| AFR | $89.0_{\pm 2.6}$ | $68.7_{\pm 1.7}$ |
| NeuronTune | $91.8_{\pm 0.8}$ | $83.0_{\pm 2.8}$ |

## A.5. Complexity Analysis

We analyze the computational complexity of our method, NeuronTune, alongside representative reweighting-based methods, including AFR (Qiu et al., 2023), DFR (Kirichenko et al., 2023), and JTT (Liu et al., 2021). Let the number of identification samples be $N_{\text{Ide}}$, the number of retraining samples be $N_{\text{ret}}$, the total number of training samples be $N$, the number of latent dimensions be $M$, and the number of training epochs be $E$. Additionally, denote the time required for inference as $\tau_{\text{fw}}$, for last-layer retraining as $\tau_{\text{ll}}$, and for optimizing the entire model as $\tau_{\text{opt}}$. The computational complexities of these methods are summarized in Table 7.

Among the methods, JTT has the highest computational complexity since $\tau_{\text{opt}} \gg \tau_{\text{ll}}$, requiring full model optimization. DFR is much faster due to its reliance on last-layer retraining, though it requires group annotations. AFR extends DFR by additionally precomputing sample losses, increasing its computational cost slightly. NeuronTune, while requiring more time than AFR to identify biased dimensions across all $M$ embedding dimensions, remains computationally efficient. This is because $\tau_{\text{fw}}$, the time required for forward inference, is typically very small. As a result, NeuronTune offers an effective balance between computational efficiency and robust spurious bias mitigation.

*Table 7.* Computation complexity comparison between NeuronTune and reweighting methods.

| Method | Time complexity |
|---|---|
| JTT (Liu et al., 2021) | $O(NE\tau_{\text{opt}})$ |
| AFR (Qiu et al., 2023) | $O(N_{\text{Ide}}\tau_{\text{fw}} + EN_{\text{ret}}E\tau_{\text{ll}})$ |
| DFR (Kirichenko et al., 2023) | $O(EN_{\text{ret}}E\tau_{\text{ll}})$ |
| NeuronTune | $O(E(N_{\text{Ide}}M\tau_{\text{fw}} + N_{\text{ret}}E\tau_{\text{ll}}))$ |

## A.6. Advantages over Variable Selection Methods

Although the identification of biased dimensions in (6) may resemble traditional variable selection methods (Heinze et al., 2018), our approach extends beyond simply selecting a subset of variables that optimally explain the target variable. Instead, it specifically addresses spurious bias—an issue often neglected in traditional variable selection.

Traditional variable selection methods, such as L1 regularization, do not differentiate between variables representing spurious attributes and those representing core attributes. Since spurious attributes are often predictive of target labels in the training data and are easier for models to learn (Tiwari & Shenoy, 2023; Ye et al., 2023), these methods may mistakenly prioritize spurious attributes, thereby amplifying spurious bias. In contrast, our method explicitly targets dimensions affected by spurious bias and rebalances the model's reliance on features to reduce dependence on spurious information.

Furthermore, unlike many variable selection methods that require explicit supervision (e.g., labels or statistical relationships) to mitigate spurious bias, NeuronTune operates in an unsupervised setting where group labels indicative of spurious attributes are unavailable. By leveraging misclassification signals to estimate spuriousness scores, our method is better suited for scenarios where group annotations are costly or infeasible, offering a practical and scalable solution to challenges of spurious bias mitigation.

## A.7. Dataset Details

Table 8 gives the details of the two image and two text datasets used in the experiments. Additionally, the ImageNet-9 dataset (Xiao et al., 2021) has 54600 and 2100 training and validation images, respectively. The ImageNet-A (Hendrycks et al., 2021) dataset has 1087 images for evaluation.

## A.8. Training Details

Table 9 and Table 10 give the hyperparameter settings for ERM and NeuronTune training, respectively.

## A.9. Visualizations of Unbiased and Biased Dimensions

We provide visualizations of the neuron activation value distributions for the identified unbiased and biased dimensions in Figs. 3 to 6. The biased and unbiased dimensions selected for visualizations are obtained by first sorting the dimensions based on their spuriousness scores and then selecting three biased dimensions that have the largest scores and three unbiased dimensions that have the smallest scores. Note that a dimension does not exclusively represent a core or spurious attribute; it typically represents a mixture of them.

On the CelebA dataset, as shown in Fig. 3, samples that highly activate the unbiased dimensions have both males and females; thus, the unbiased dimensions do not appear to have gender bias. For samples that highly activate the identified biased dimensions, all of them are females, demonstrating a strong reliance on the gender information. In Fig. 4, samples that highly activate the identified biased dimensions (right side of Fig. 4) tend to have slightly darker hair colors or backgrounds, as compared with samples that highly activate the identified unbiased dimensions (left side of Fig. 4). With the aid of the heatmaps, we observe that these biased dimensions mostly represent a person's face, which is irrelevant to target classes.

On the Waterbirds dataset, as shown in Fig. 5, for the landbird class, the identified unbiased dimensions mainly represent certain features of a bird and land backgrounds. For the identified biased dimensions, they mainly represent water backgrounds, which are irrelevant to the landbird class based on the training data. For the waterbird class, as shown in Fig. 6, the identified unbiased dimensions mostly represent certain features of a bird and water backgrounds, while the identified biased dimensions mainly represent land backgrounds.

*Table 8.* Numbers of samples in different groups and different splits of the four datasets.

| Class | Spurious attribute | Train | Val | Test |
|---|---|---|---|---|
| Waterbirds | | | | |
| landbird | land | 3498 | 467 | 2225 |
| landbird | water | 184 | 466 | 2225 |
| waterbird | land | 56 | 133 | 642 |
| waterbird | water | 1057 | 133 | 642 |
| CelebA | | | | |
| non-blond | female | 71629 | 8535 | 9767 |
| non-blond | male | 66874 | 8276 | 7535 |
| blond | female | 22880 | 2874 | 2480 |
| blond | male | 1387 | 182 | 180 |
| MultiNLI | | | | |
| contradiction | no negation | 57498 | 22814 | 34597 |
| contradiction | negation | 11158 | 4634 | 6655 |
| entailment | no negation | 67376 | 26949 | 40496 |
| entailment | negation | 1521 | 613 | 886 |
| neither | no negation | 66630 | 26655 | 39930 |
| neither | negation | 1992 | 797 | 1148 |
| CivilComments | | | | |
| neutral | no identity | 148186 | 25159 | 74780 |
| neutral | identity | 90337 | 14966 | 43778 |
| toxic | no identity | 12731 | 2111 | 6455 |
| toxic | identity | 17784 | 2944 | 8769 |

*Table 9.* Hyperparameters for ERM training.

| Hyperparameters | Waterbirds | CelebA | ImageNet-9 | MultiNLI | CivilComments |
|---|---|---|---|---|---|
| Initial learning rate | 3e-3 | 3e-3 | 1e-3 | 1e-5 | 1e-3 |
| Number of epochs | 100 | 20 | 120 | 10 | 10 |
| Learning rate scheduler | CosineAnnealing | CosineAnnealing | MultiStep[40,60,80] | Linear | Linear |
| Optimizer | SGD | SGD | SGD | AdamW | AdamW |
| Backbone | ResNet50 | ResNet50 | ResNet18 | BERT | BERT |
| Weight decay | 1e-4 | 1e-4 | 1e-4 | 1e-4 | 1e-4 |
| Batch size | 32 | 128 | 128 | 16 | 16 |

*Table 10.* Hyperparameters for NeuronTune.

| Hyperparameters | Waterbirds | CelebA | ImageNet-9 | MultiNLI | CivilComments |
|---|---|---|---|---|---|
| Learning rate | 1e-3 | 1e-3 | 1e-3 | 1e-5 | 1e-3 |
| Number of batches per epoch | 200 | 200 | 200 | 200 | 200 |
| Number of epochs | 40 | 40 | 1 | 60 | 60 |
| Optimizer | SGD | SGD | SGD | AdamW | AdamW |
| Batch size | 128 | 128 | 128 | 128 | 128 |

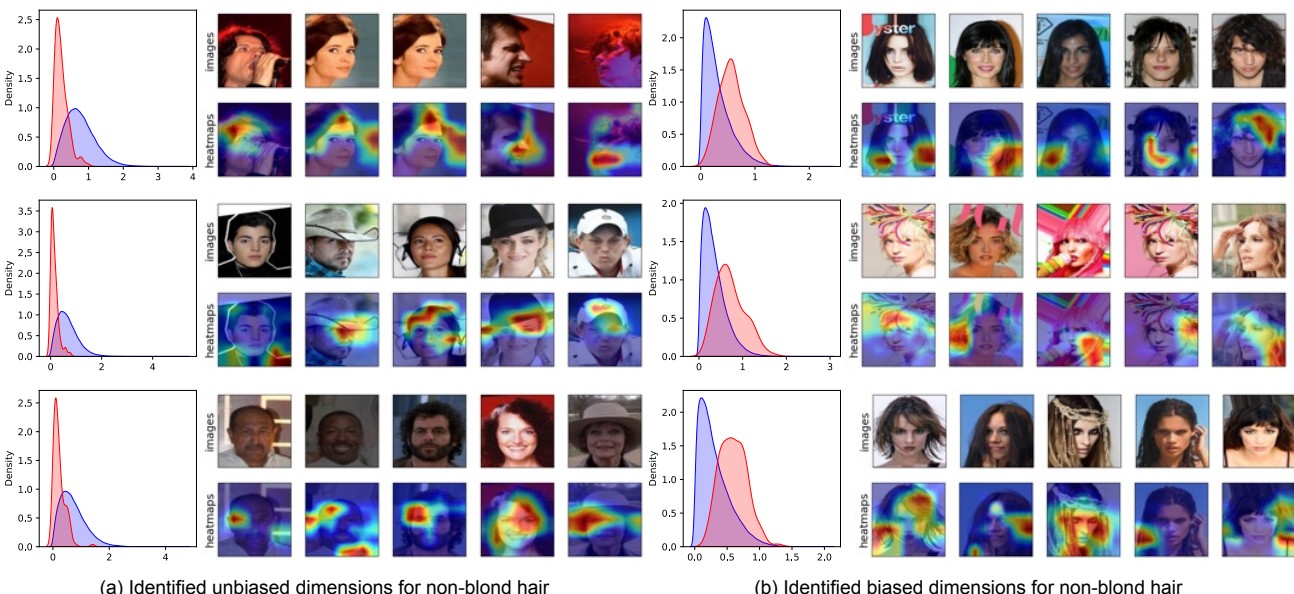

(a) Identified unbiased dimensions for non-blond hair

(b) Identified biased dimensions for non-blond hair

*Figure 3.* Value distributions of the correctly (blue) and incorrectly (red) predicted samples for unbiased (a) and biased (b) dimensions, along with the representative samples, respectively, based on the non-blond hair samples in the CelebA dataset.

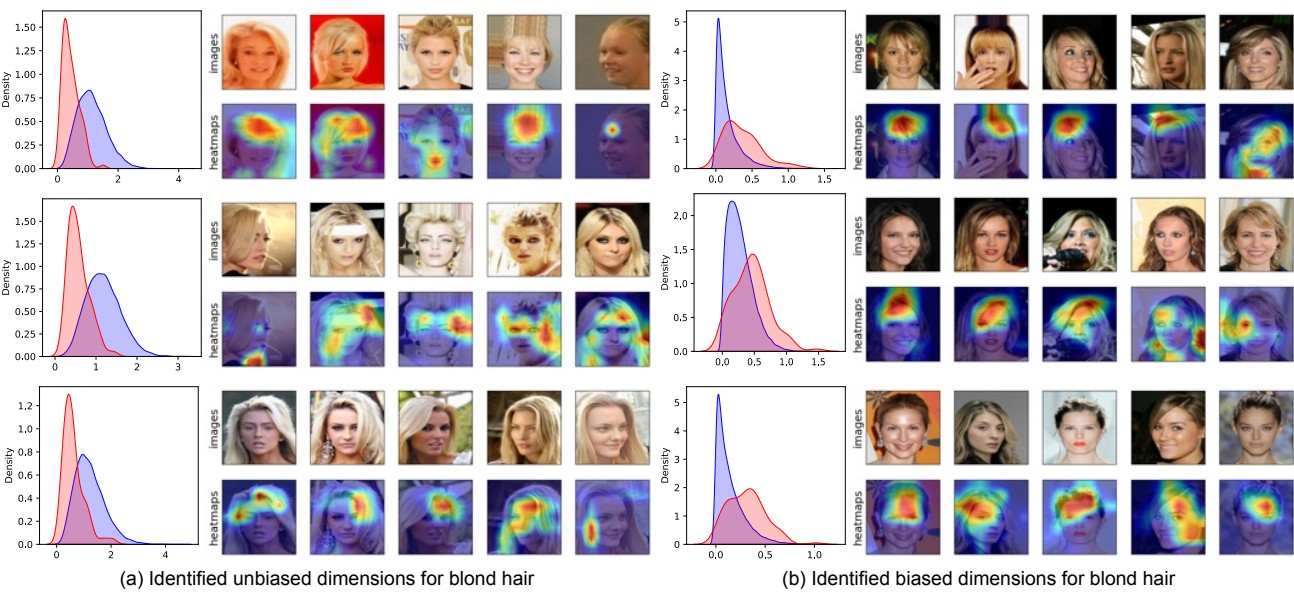

(a) Identified unbiased dimensions for blond hair

(b) Identified biased dimensions for blond hair

*Figure 4.* Value distributions of the correctly (blue) and incorrectly (red) predicted samples for unbiased (a) and biased (b) dimensions, along with the representative samples, respectively, based on the blond hair samples in the CelebA dataset.

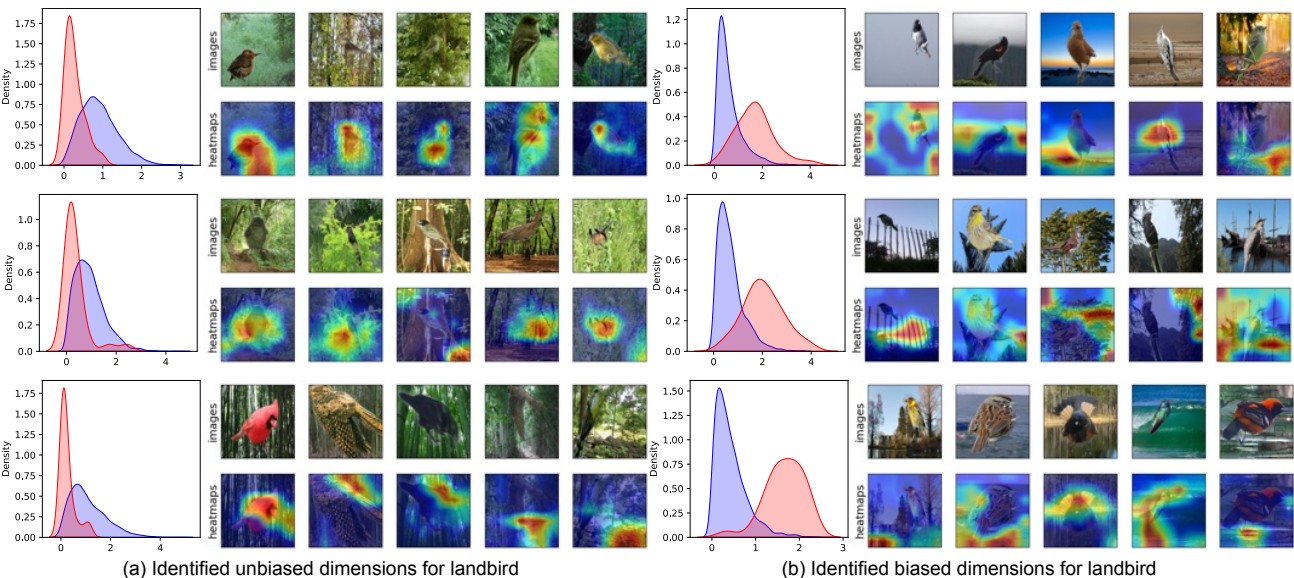

*Figure 5.* Value distributions of the correctly (blue) and incorrectly (red) predicted samples for unbiased (a) and biased (b) dimensions, along with the representative samples, respectively, based on the landbird samples in the Waterbirds dataset.

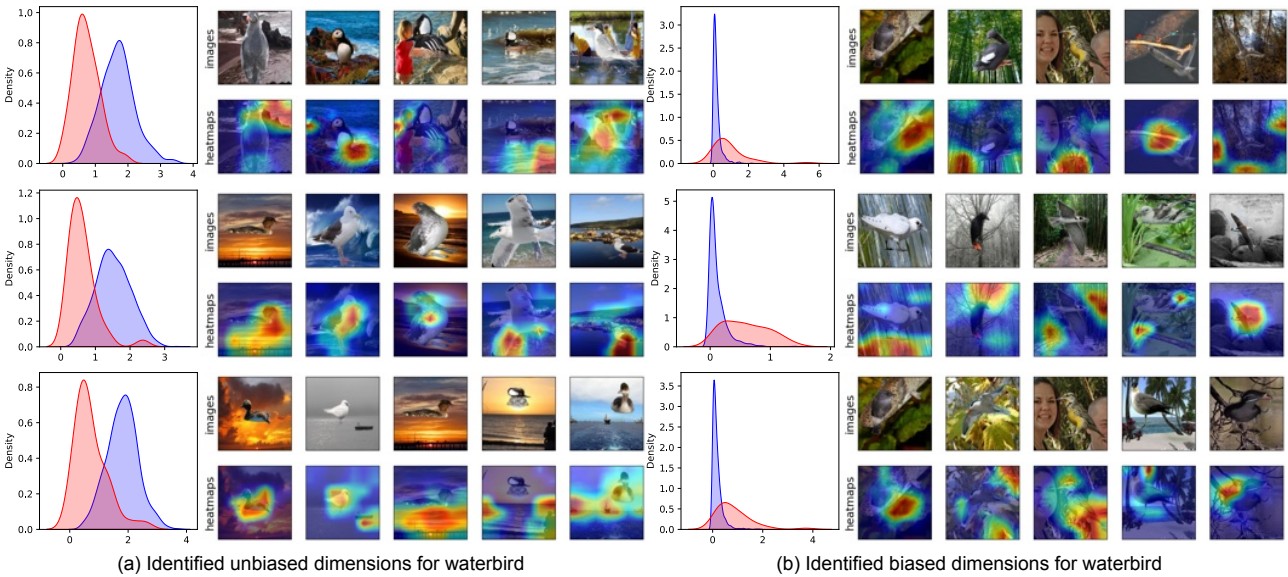

*Figure 6.* Value distributions of the correctly (blue) and incorrectly (red) predicted samples for unbiased (a) and biased (b) dimensions, along with the representative samples, respectively, based on the waterbird samples in the Waterbirds dataset.

