# OpenReview forum: "NeuronTune: Towards Self-Guided Spurious Bias Mitigation"
_ICML.cc/2025/Conference — ICML 2025 poster_

### Official Review · Reviewer_fYLW · 2025-02-19

**Overall Recommendation:** 4

**Summary:**

The paper proposes NeuronTune, a method for mitigating spurious correlations in neural networks by intervening directly in the latent space at the neuron (feature-dimension) level. The work addresses the common challenge that many “robustness” or “debiasing” approaches rely on knowing or inferring spurious attribute annotations (e.g., “group labels”), which can be expensive to obtain or sometimes even unknown a priori. NeuronTune provides a self-guided approach: it identifies “biased dimensions” whose activations correlate strongly with misclassifications and suppresses those dimensions through post-hoc last-layer retraining.

**Claims And Evidence:**

Overall, the claims are backed by both theoretical insights and strong empirical evidence.

**Essential References Not Discussed:**

A potentially relevant reference is “NeuroInspect,” which also performs neuron-level interventions. Referencing it explicitly could strengthen the discussion on interpretable neuron-level debugging strategies.

**Experimental Designs Or Analyses:**

The methodology is generally rigorous. One natural trade-off is a mild drop in average accuracy for a bigger gain in worst-group performance.

**Methods And Evaluation Criteria:**

These criteria align with standard practices in the literature on spurious correlation mitigation.

**Other Comments Or Suggestions:**

- Iterative neuron identification: After carrying out one pass of NeuronTune, it might be worthwhile to re-run the biased-neuron detection on the newly tuned model. If certain features remain entangled or if new spurious signals emerge, a second iteration could potentially refine results.
- Interpretation: It would be interesting to present visual or textual exemplars that highlight what the suppressed neurons correspond to semantically.

**Other Strengths And Weaknesses:**

## Strengths:
- Self-contained, requiring no group-label supervision.
- Effective across various architectures (ResNet, BERT) and data modalities (vision and NLP).
- Theoretical rationale matches empirical success.

## Weaknesses:
- Limited interpretability: Although the approach identifies “biased neurons,” the paper does not deeply explore how to interpret or label them. (This is not a critical flaw, but might be interesting for users to see “which spurious concepts are discovered.”)
- Dependence on a separate identification set: The paper relies heavily on the notion that a distinct dataset (or at least a validation set) is available. In real scenarios, that might be limited or small. The ablations show that using training data to identify biased dimensions is less effective (Table 4).

**Questions For Authors:**

1. Iterative Tuning: Have you explored running multiple rounds of neuron identification and suppression? If so, did you notice any diminishing returns or additional complexity? If not, do you speculate that repeated rounds might identify new “biased” neurons or further improve performance?

2. Partial vs. Full Suppression: Did you consider only attenuating (rather than fully zeroing) the activations of biased dimensions? Could such a variant reduce potential losses in overall accuracy?

3. Identification Set Size/Composition: Are there heuristics or rules of thumb for how large or diverse the identification set (used to detect biased neurons) should be? For instance, if only limited validation data is available, how does that affect the stability of the median-based criterion?

4. Interpretability: Have you tried mapping the suppressed neurons to specific “concepts” or features (e.g., through visualization)? If so, does this reveal anything interesting about the nature of spurious correlations the model relies upon?

5. Extension to Other Tasks: The paper focuses on classification tasks with known spurious correlations. Could the NeuronTune concept extend to multi-label or structured prediction tasks? If so, would that require any modification to the methodology?

**Relation To Broader Scientific Literature:**

The authors contextualize well with respect to spurious bias research and last-layer retraining. They might also elaborate on recent neuron-level debugging works such as “NeuroInspect: Interpretable Neuron-based Debugging Framework through Class-conditional Visualizations” [Ju et al., 2023] to compare or contrast how each approach manipulates internal neurons.

**Theoretical Claims:**

Key theoretical elements include:

1. A formal data model that encapsulates both core and spurious features.
2. Proofs establishing the connection between median-activation differences and biased (spurious) neuron detection.
3. Analysis that shows last-layer retraining plus biased-neuron suppression reduces reliance on spurious features.

The analysis is coherent for linearized settings and helps explain why high-activation, misclassification-associated neurons reveal spurious correlations.

---

> ### Author Rebuttal · Authors · 2025-03-31
>
> Thank you for your detailed review and feedback on our submission. We provide our responses to your questions below.
>
> ## Essential References Not Discussed
> - **Comparison with NeuroInspect**: NeuroInspect identifies neurons responsible for mistakes from the counterfactual explanation perspective and minimizes the influence of identified neurons via regularization. In contrast, our approach identifies biased neurons by probing their activation patterns and directly manipulates identified neurons by suppressing their contributions to the final predictions.\
> NeuroInspect emphasizes on interpretable neuron-level debugging, allowing practitioners to further pinpoint the causes of the errors. Our work aims at a self-guided spurious bias detection and mitigation framework without human intervention.\
> We see potential in applying the interpretation techniques to our method in future work. We will cite NeuroInspect and add the above discussion to our revised manuscript.
>
> ## Weaknesses
>
> 1. **Limited interpretability**: We hope to clarify that our work does not focus on interpretation. Instead, we focus on a self-guided framework without human intervention. By definition, biased neurons may encode a certain portion of core features, making the interpretation of neurons nontrivial. Nevertheless, we provide visualizations in Figures 3-6 in Appendix to analyze which spurious concepts are discovered. Please refer to our response to Question 4 for details.
>
> 2. **Dependence on a separate identification set**: One of our major contributions is removing the dependence on group labels in the identification set. This offers great flexibility and freedom for practitioners to select identification data based on their needs to detect and mitigate targeted spurious biases. Hence, obtaining such data is relatively straightforward. In our experiments, we used the validation set for consistency in performance comparison, but in real-world scenarios, any sufficiently representative data can serve this purpose.
>
> ## Questions:
>
> 1. **Iterative Tuning**: Yes, we performed multiple rounds of neuron identification and suppression (L256-259, right column). Diminishing returns will be observed after a certain number of rounds. In general, running multiple rounds is better than running a single round with improved performance, as new biased neurons will be identified.
>
> 2. **Partial vs. Full Suppression**:  Although partial suppression reduces the loss in overall accuracy, its effect is similar to no suppression, as models can continue to adjust their weights on nonzero biased activations, leading to severe spurious bias. For example, on the CelebA dataset, when we multiplied the activations of biased dimensions with the masking values in the table below, we see that only when the masking value is zero (full suppression)  did the model achieve improved WGA.
>
> |Masking value|0|0.2|0.4|0.6|0.8|1.0|
> |---|---|---|---|---|---|---|
> |WGA|87.3±0.4|71.5±1.5|72.2±1.2|72.9±1.5|73.1±1.5|73.0±1.2|
> |Acc.|90.3±0.5|93.8±0.2|93.8±0.3|93.8±0.2|93.8±0.2|93.9±0.2|
>
> 3. **Identification Set Size/Composition**: The identification set should consist of diverse samples that are not memorized by the model. We find that when we use less than 30\% samples of the validation set, the obtained worst-group accuracy is low with high variance, whereas the average accuracy remains high with low variance. When we use around 30\% of the validation set, the number of minority group samples (e.g., waterbirds with land backgrounds in Waterbirds or males with blonde hair in CelebA) is approximately 50. Therefore, to ensure the effectiveness of our method, each data group should ideally contain more than 50 samples. In a group-label-free setting, it is suggested to include many diverse samples whenever possible.
>
> 4. **Interpretability**: Yes, we have shown images that highly activate the identified biased neurons along with the heatmaps highlighting regions that contribute to the neuron activations in Figures 3-6 in Appendix. The identified biased neurons mainly represent spurious features. For example, we observe that biased neurons mainly represent blue/green backgrounds for landbirds in Figure 5(b) and represent forest backgrounds for waterbirds in Figure 6(b). Note that since a neuron may represent a mixture of spurious and core features, the visualizations sometimes may not be directly interpretable.
>
> 5. **Extension to Other Tasks**: Yes, NeuronTune can be extended to multi-label prediction tasks. Instead of considering all identified neurons for different labels as biased, as shown in Eq. (6), we can consider biased neurons as label-specific and mask neurons based on the labels of input samples during retraining. This is because in the multi-label setting, a biased neuron for one label may be the core contributor to another label, which is different from the assumption in a standard classification setting (L272-274).
>
> Please let us know if you have further questions.

---

### Official Review · Reviewer_MxxR · 2025-03-13

**Overall Recommendation:** 3

**Summary:**

The authors propose to improve OOD generalization by identifying neurons that contribute significantly to misclassification on the validation set, and then retraining the output layer while setting those neurons to zero. Relative to ERM, this simple idea trades off in-distribution accuracy for significant improvement on worst-group accuracy across many image and text classification tasks.

**Claims And Evidence:**

* The authors' main claim is that their algorithm is robust to "spurious correlations," which are defined as a non-causal relationship.
* Their approach is motivated using a DGP with causal/anticausal features, originally proposed in the Invariant Causal Prediction (Peters et al., 2015) paper.
* Their experimental results show that their algorithm is able to effectively trade-off in-distribution performance for out-of-distribution performance in the presence of subpopulation shift.
* I believe the connection between the claims and evidence is currently weak, and would be improved by reframing the algorithm as a way to mitigate subpopulation shift.

**Essential References Not Discussed:**

N/A

**Experimental Designs Or Analyses:**

* The experimental design and analysis are sound.
* I liked the fact that the authors are very explicit about how they perform model selection using in-distribution data. This is an extremely important point that is often omitted in this research area.

**Methods And Evaluation Criteria:**

* Yes, the proposed methods and evaluation make sense for the problem of OOD generalization.

**Other Comments Or Suggestions:**

N/A

**Other Strengths And Weaknesses:**

The proposed idea is elegant, and meaningfully beats ERM on a wide variety of very difficult benchmarks. However, I think the current causality-based presentation obfuscates the simplicity of the idea, and does not help the reader gain intuition on why it should improve OOD generalization.

This is how I understand your method. When you train ERM on datasets with imbalanced subpopulations, it maximizes average-case accuracy and performs very poorly on rare subpopulations. E.g. on waterbirds, it focuses on being correct on waterbirds on water, and sacrifices its ability to classify waterbirds on land. Then, when you look at the misclassifications on the validation set, the features that you identify are precisely the ones that make you misclassify waterbirds on land. You zero these features out, and retrain the output layer to assign higher weights to the features that would allow you to correctly classify waterbirds on land, i.e. bird features. By doing this, you slightly sacrifice your ability to classify waterbirds on water, but significantly improve your ability to classify waterbirds on land. Hence, your algorithm improves on worst-group accuracy at the expense of average-case accuracy. The same conclusion can be reached using CivilComments, which is categorized as a subpopulation shift dataset within WILDS.

I think this makes much more sense than the causality-based argument which hinges on unintuitive technical assumptions about the data generating process. I am going to recommend acceptance as-is, because I understand that my interpretation may be overfit to my own personal intuitions about this research area. However, I believe this paper would be stronger with this reframing.

**Questions For Authors:**

Let me know what you think about the subpopulation shift interpretation that I proposed in "other strengths and weaknesses."

**Relation To Broader Scientific Literature:**

The authors primarily contextualize their work in relation to last-layer retraining with labeled spurious correlations, and domain generalization (labeled groups). I think this is fine, but I believe the work more strongly relates to subpopulation shift (see "Other strengths and weaknesses.")

**Theoretical Claims:**

N/A

---

> ### Author Rebuttal · Authors · 2025-03-31
>
> Thank you for your thoughtful comments and for recognizing both the simplicity and effectiveness of our approach. We appreciate your valuable feedback and are glad that our efforts to explain the model selection strategies were helpful.
>
> ## Regarding the subpopulation shift interpretation
>
> We appreciate your thorough understanding of our work and your insightful interpretation from the perspective of subpopulation shifts, i.e., changes in the proportion of certain subpopulations between training and testing. We would also like to clarify the distinction between mitigating **spurious bias** and **subpopulation shifts**.
>
> - Mitigating subpopulation shifts often relies on reducing spurious bias, such as decreasing the strong correlation between waterbirds and water to improve generalization for waterbirds on land. Tables 1 and 2 demonstrate NeuronTune’s effectiveness in handling subpopulation shifts on the Waterbirds, CelebA, MultiNLI, and CivilComments datasets.
>
> - In contrast, mitigating spurious bias has broader implications beyond subpopulation shifts. For example, Table 3 highlights NeuronTune’s ability to enhance robustness on the ImageNet-A dataset, which consists of samples that are challenging for a pre-trained model but do not conform to well-defined subpopulation shifts, such as images with unique pixels. This evaluation scenario represents a broader and more complex problem than subpopulation shifts.
>
> **Therefore, by framing our method as targeting spurious bias, we address a wider scope of challenges beyond subpopulation shifts.**
>
> With proper examples,  our current presentation aligns well with your proposed subpopulation shift interpretation. We detail the correspondence between your interpretation and our presentation below.
>
> > When you train ERM on datasets with imbalanced subpopulations, it maximizes average-case accuracy and performs very poorly on rare subpopulations. E.g. on waterbirds, it focuses on being correct on waterbirds on water, and sacrifices its ability to classify waterbirds on land.
>
> Consider that $a$ in Eq. (2) controls subpopulations in data, e.g., when $a=1$, it may represent a group of waterbirds on water, and when $a=0$, it may represent a group of waterbirds on land. The probability $p$ controls the severity of imbalance in subpopulations. When $p$ is close to one (L159 left), the data is severely imbalanced in subpopulations. After training with ERM, the model minimizes the training loss, i.e., maximizes average-case accuracy, but obtains a large nonzero weight on the spurious feature (Lemma 1) and is away from the optimal model (Corollary 1). For example, the model may focus on being correct on waterbirds on water and sacrifice its ability to classify waterbirds on land.
>
> > Then, when you look at the misclassifications on the validation set, the features that you identify are precisely the ones that make you misclassify waterbirds on land. You zero these features out, and retrain the output layer to assign higher weights to the features that would allow you to correctly classify waterbirds on land, i.e., bird features.
>
> Our principle of neuron identification (Proposition 4.1) states that when a spurious correlation breaks, the neurons that still positively contribute to mispredictions will be selected.  For example, neurons that lead to misclassification on waterbirds on land will be identified. After retraining, Theorem 4.3 indicates that weights on core features, such as bird features, will be increased (L200-201, right).
>
> > By doing this, you slightly sacrifice your ability to classify waterbirds on water, but significantly improve your ability to classify waterbirds on land.
>
> Our findings (L196-201, right) show that our method increases the weights on core features while keeping the weights on spurious features unchanged. This suggests that our approach makes a slight trade-off in average-case accuracy to achieve improved worst-group accuracy.
> For example, our method may slightly reduce the model’s ability to classify waterbirds on water due to a relative decrease in reliance on the water feature, while significantly enhancing its ability to classify waterbirds on land.
>
>  Thanks again for your thoughtful comments. We will incorporate concrete examples to provide readers with better intuition and a clearer understanding of our approach.
>
> Please let us know if you have further questions.

---

### Official Review · Reviewer_BqZa · 2025-03-15

**Overall Recommendation:** 4

**Summary:**

The work proposes a bias-unaware post-hoc model debiasing method. The approach
is based on the observation that, in a setting where a high majority of
samples, but not all, contain a biased attribute, the neurons in the
penultimate layer that are affected by the spurious attribute exhibit a
different behavior between correctly classified and misclassified samples.
The work theoretically motivates this in a linear regression setting, and
defines the difference between medians of classified and misclassified samples
as an indicator for neurons that are sensitive to spurious attributes.
The debiasing is conducted by setting neurons in the penultimate layer above a
certain distance threshold to zero, and retraining the final classification
layer. A toy example serves to further highlight the efficacy of this approach.
Empirical experiments on four popular benchmark datasets show
worst-group-accuracy values of the proposed method competitive with approaches
from prior work which leverage bias labels in the validation set for model
selection.

## update after rebuttal

I thank the authors for their clarifications. I am very happy to see the increased number of trials. I am convinced of the quality, novelty and significance of this work, and thus choose to revise my initial recommendation to "Accept".

**Claims And Evidence:**

The work claims their proposed debiasing approach, which does not require bias
annotations, significantly mitigates spurious biases, i.e., is competitive with
previous bias mitigation approaches that do require bias annotations.
This claim is supported by empirical evidence, comparing the
worst-group-accuracy of various baselines. There seems to be a small issue given
the high error, likely caused by the small number of conducted trials. However,
as the work does not explicitly claim state-of-the-art, but only
competitiveness, this is only a minor issue.

**Essential References Not Discussed:**

This works bases its bias detection on the idea that biased models misclassify
bias-conflicting samples. Two prior works that leverage the same idea are cited
and compared empirically (Kim et al, 2022a; Liu et al., 2021). However, the work
does not make this connection clear.

**Experimental Designs Or Analyses:**

The experimental design of the results discussed on Tables 1-3 is based on a
straight-forward comparison on previous benchmarks. The only potential issue
here could be the low number for trials (three).

**Methods And Evaluation Criteria:**

The work conducts empirical analysis on common benchmark datasets from the model
debiasing literature. Here, the worst-group performance is used as an evaluation
criteria, which is a common metric.
The work also uses the *accuracy gap* to evaluate the gain between average and
worst-group-performance. While I understand the motivation behind this metric, I
do not think that this is very useful, as models with very low
worst-group-performance obtain inflated values, which still require the
comparison to average and worst-group-performance.

The work also proposes a synthetic dataset to visually motivate its approach.
While the visualization does indeed provide a good intuition for the issue.

**Other Comments Or Suggestions:**

- In Figure 2b: The median for Dimension 2 red seems off.

- l.255 left: linear combination spurious and core in embedding typically holds in embedding space, "combination is nonlinear" should be specified more clearly (i.e., in input)
- l.307 left: absolution -> absolute

- l.791: so that suppress[ing] them improves the ...

**Other Strengths And Weaknesses:**

### Strengths

- Unsupervised bias mitigation, not relying on known biases, is an interesting
  and important issue
- The linear regression example motivates the approach well. I really like
  Figure 2, except for the issue of the decision boundary described below.
- The theoretical analysis provides a good foundation for the efficacy of the
  model.
- The work is written clearly, and seems polished well.

### Weaknesses

- The empirical results have high error due to a low number of trials, and do
  not show very clear improvements.
- The framework relies on the assumption that a overwhelming majority of the
  training set contains biased samples in order for the model to react to
  bias-conflicting samples.

**Questions For Authors:**

1. The error values in Table 2 are quite high. Could this be caused by the low
   number of trials? Would at least e.g. five trials be possible? Did you
   conduct a Wilcoxon-rank-sum test?
2. The decision boundaries are different between training and testing in Figure
   2a and 2c. Is this caused by errors in the contour lines? Should they not be
   exactly the same?
3. l.379 left states "Our method achives highest WGA across the datasets" for
   Table 1 and Table 2. However, this does not seem to be fully the case. E.g.,
   On Multi NLI, AFR is just overall better, yet NeuroTune is in bold-face.
4. When using the validation set to find neurons that are sensitive to spurious
   behavior, is the training set used to tune the last layer?

**Relation To Broader Scientific Literature:**

The work discusses most relevant works, making the common separation between
bias-aware, bias-unaware, and semi-bias-aware mitigation approaches.
The idea that bias can be identified through the misclassification of
bias-conflicting samples has been identified in prior work (Kim et al, 2022a;
Liu et al., 2021).
Reweighing or retraining the final layer for bias mitigation has also been
leveraged in previous work (Liu et al., 2021; Qiu et al., 2023).
The novelty in this work is the contribution of individual neurons to this
misclassification, and the subsequent, targeted removal thereof.

**Theoretical Claims:**

The work makes some theoretical claims on a linear regression model in order to
motivate its method. Here, Proposition 4.1, given separated spurious and core
features in a linear model, demonstrates the negative contribution of spurious
neurons.

Given these negative contributions, Theorem 4.2 shows that there is then some
distance between the expected activations of correctly classified and misclassified samples relying on this negative contribution of Proposition 4.1.

Theorem 4.3 then assumes that the neurons detecting core features are aligned
with the true core direction approximately to the same degree as the neurons
detecting spurious features are aligned to the true spurious direction, with
which the proposed approach is shown to reduce the distance between the unbiased
and the biased model.

I did not rigorously check Theorems 4.2 and 4.3.

---

> ### Author Rebuttal · Authors · 2025-03-31
>
> Thank you for your thoughtful review and valuable feedback on our submission. Here are our responses to address your questions.
>
> ## Evaluation Criteria
> - **Usefulness of Accuracy Gap**: We hope to explicitly provide readers with a direct view on the numeric difference between worst-group accuracy and average accuracy, making it easier to assess how consistently a model performs across different groups of data. When models with very low worst-group accuracy exhibit inflated accuracy gaps, such as the ERM models, this indeed highlights the severe robustness issue of the models. In this paper, we report the average accuracy, worst-group accuracy, and the accuracy gap between them to provide a comprehensive evaluation of each model's overall performance.
>
> ## Essential References Not Discussed
> - **Connection to the two related works**: Thanks to the reviewer for giving a clear comparison between our work and the works proposed by Kim et al, 2022a and Liu et al., 2021. Indeed, our work builds on idea of these works by extending the focus from identifying bias-conflicting samples to detecting biased neurons within a model, enabling direct intervention in the model’s decision-making process.
> We will add the above discussion to our revised manuscript to better connect our method to these two works.
>
> ## Weaknesses
>
> 1. **Regarding number of trials and clear improvements**:  Following your suggestion, we ran our method for five additional trials. Overall, the accuracies remained stable with reduced variance. Our previous results in the paper and the new results below clearly demonstrate the advantage of our method in the unsupervised bias mitigation setting, where no group labels are available.
>
> || NeuronTune ||NeuronTune$^\dagger$||
> |---|---|---|---|---|
> ||WGA|Acc.|WGA|Acc.|
> |Waterbirds|92.2±0.3|94.4±0.2|92.5±0.9|94.5±0.3|
> |CelebA|83.1±1.1|92.0±0.5|87.3±0.4|90.3±0.5|
> |MultiNLI|72.1±0.1|81.1±0.6|72.5±0.3|80.3±0.6|
> |CivilComments|82.4±0.2|89.2±0.1|82.7±0.4|89.4±0.2|
>
> 2. **Regarding the assumption on the training set**:  We would like to clarify that our method does not rely on any assumptions about the training set. Since our approach uses an independent identification set for bias detection, it can be applied to models trained on any datasets. Notably, if the training set contains fewer biased samples, the resulting model is less susceptible to spurious bias. In such cases, NeuronTune will identify fewer biased neurons and the model will receive less intervention for bias mitigation.
>
> ## Other Comments Or Suggestions
> We will make revisions at the suggested locations and thoroughly review the rest of our paper.
>
> ## Questions
>
> 1. **The error values in Table 2 are quite high. Could this be caused by the low number of trials? Would at least e.g. five trials be possible? Did you conduct a Wilcoxon-rank-sum test?** We hope to clarify that the variances of our method are relatively low compared to other methods. After running five additional trials, we observed reduced variance. Please refer to the table in Weakness 1 for the results. Our method demonstrates clear improvements in the unsupervised bias mitigation setting. Since we compare against the best-reported performance of baseline methods, a Wilcoxon rank-sum test is not directly applicable in this context.
>
> 2. **The decision boundaries are different between training and testing in Figure 2a and 2c**:  This discrepancy arises because we used two separate K-nearest neighbor classifiers, each fitted to the predictions on the training and test sets separately, to determine the decision boundaries. To ensure consistency, we have now used a shared K-nearest neighbor classifier fitted on both training and test data to determine decision boundaries. The revised figure can be accessed  [here](https://anonymous.4open.science/r/NeuronTune-6CFC/synthetic_example.png).
>
> 3. **Regarding the statement in l.379 left**: We hope to clarify that in Tables 1 and 2, we group methods that are directly comparable and boldface the best result within each group of methods (L345 and L398). For example, on MultiNLI (Table 2), AFR belongs to the methods that require group labels, while NeuronTune does not require group labels.
> NeuronTune achieves the highest WGA within methods that do not require group labels and even remains competitive against methods that require group labels. We will revise L379 to further clarify this in the paper.
>
> 6. **When using the validation set to find neurons that are sensitive to spurious behavior, is the training set used to tune the last layer?** Yes, for "NeuronTune," the validation set is used only to identify biased neurons, while the training set is reused to tune the last layer. "NeuronTune$^\dagger$" is a variant where half of the validation set is used for detection and the other half for tuning.
>
>
>  Please let us know if you have further questions.

---

### Decision · Program_Chairs · 2025-05-01

**Decision:**

Accept (poster)

**Comment:**

This paper received three Weak Accept recommendations.

The reviewers acknowledged the theoretical analysis that serves as a foundation for the proposed approach, its novelty in terms of the identification of individual neurons affected by bias, suppressing their contributions for bias mitigation, and experiments including CNNs and BERT on both image and text classification tasks.  On the other hand, reviewers raised concerns about the assumption that the vast majority of the training samples are bias-aligned, low number of trials, connection between claims and provided experimental evidences, more in-depth analysis on the identified biased neurons as well as the dependence on a validation set, which may be hard to obtain in a realistic scenario.

In their rebuttal, the authors clarify how they do not use group labels in the validation set, they perform more trials to provide more indication on the method’s robustness, as well as clarifying that the proposed method does not assume a specific bias-aligned/bias-conflicting ratio in the training set, but would likely find less biased with the final model receiving less intervention, and provide more details on the setting and design choices.

After reading the paper, the AC has a further comment on the related work section, specifically in the unsupervised Spurious Bias Mitigation subsection, which appears not articulated enough, as it only reports one work and misses significant contributions in this field. Especially as the paper fits into this bias mitigation framework, describing and comparing against more of these works would be useful to assess the performance of the proposed approach correctly.

Considering the reviewers’ scores and carefully evaluating the strengths and weaknesses of the proposed approach, the AC agrees that the proposed method is interesting and novel in its identification of biased neurons, with experiments showing its effectiveness. Therefore, they recommend the acceptance of this paper.